# Analysis of off-tumour toxicities of T-cell-engaging bispecific antibodies via donor-matched intestinal organoids and tumouroids

Marius F. Harter[1,2], Timothy Recaldin[3], Regine Gerard[3], Blandine Avignon[3], Yannik Bollen[1], Cinzia Esposito[4], Karolina Guja-Jarosz[4], Kristina Kromer[3], Adrian Filip[1], Julien Aubert[1], Anneliese Schneider[5], Marina Bacac[5], Michael Bscheider[3], Nadine Stokar-Regenscheit [3], Salvatore Piscuoglio[4], Joep Beumer[1] & Nikolche Gjorevski [1]✉

Predicting the toxicity of cancer immunotherapies preclinically is challenging because models of tumours and healthy organs do not typically fully recapitulate the expression of relevant human antigens. Here we show that patient-derived intestinal organoids and tumouroids supplemented with immune cells can be used to study the on-target off-tumour toxicities of T-cell-engaging bispecific antibodies (TCBs), and to capture clinical toxicities not predicted by conventional tissue-based models as well as inter-patient variabilities in TCB responses. We analysed the mechanisms of T-cell-mediated damage of neoplastic and donor-matched healthy epithelia at a single-cell resolution using multiplexed immunofluorescence. We found that TCBs that target the epithelial cell-adhesion molecule led to apoptosis in healthy organoids in accordance with clinical observations, and that apoptosis is associated with T-cell activation, cytokine release and intra-epithelial T-cell infiltration. Conversely, tumour organoids were more resistant to damage, probably owing to a reduced efficiency of T-cell infiltration within the epithelium. Patient-derived intestinal organoids can aid the study of immune–epithelial interactions as well as the preclinical and clinical development of cancer immunotherapies.

Cancer immunotherapy, which recruits immune cells in targeting tumours, has emerged as one of the most promising strategies for battling cancer[1,2]. However, despite instances of stunning clinical success[3,4], only a fraction of patients and tumour subsets respond owing to a confluence of factors, including a low expression of neoantigens by cancer cells and immunosuppressive tumour milieus[5]. In addition to inconsistent antitumour efficacy, immune-related toxicities are a major roadblock to the clinical translation of cancer-immunotherapy drugs. The enormous clinical potential of nearly all T cell-targeted approaches, including immune checkpoint inhibitors, chimaeric antigen receptor (CAR) T cells and bispecific T-cell engagers, is undercut by on-target activity in healthy tissues, ultimately resulting in serious adverse

[1]Institute of Human Biology (IHB), Roche Innovation Center Basel, Basel, Switzerland. [2]Gustave Roussy Cancer Campus, University Paris-Saclay, Paris, France. [3]Roche Pharma Research and Early Development, Roche Innovation Center Basel, Basel, Switzerland. [4]Department of Biomedicine, University of Basel, Basel, Switzerland. [5]Roche Pharma Research and Early Development, Roche Innovation Center Zurich, Schlieren, Switzerland. ✉e-mail: nikolche.gjorevski@roche.com

effects[6–8]. These toxicities, ranging from cytokine release syndrome to organ damage, often lead to the termination of promising new clinical programs and limit the broad application of approved therapies[9–11]. However, most of these challenges are typically unforeseen by traditional preclinical toxicology models, including cell lines and animals, which either fall short in capturing the complexity of native organs, or lack human-specific tissue features and immunological responses[12,13].

Patient-derived organoids—three-dimensional (3D) structures derived from primary healthy or tumour tissue samples—are coming of age as powerful preclinical models owing to their ability to preserve morphological, genetic and functional features of the parental tissue[14,15]. The value of organoids in oncology research and personalized medicine has been exemplified by studies demonstrating accurate prediction of patient responses to anticancer drugs[16–19]. Although organoids themselves are devoid of an immune compartment, supplementation with immune cells has enabled their use for preclinical testing of not only chemotherapies or targeted therapies but also immunotherapies[18,20]. These studies have provided compelling evidence that organoids can be used to test and improve the efficacy of cancer-immunotherapy drugs and tailor their clinical application to patients with a high chance of response. However, organoid-based immuno-oncology models have largely overlooked toxicity of cancer immunotherapy in healthy tissues as a major hurdle in the effective clinical translation of these drugs.

In this study, we applied healthy intestinal organoids to evaluate the safety liabilities of cancer-immunotherapy drugs, using T-cell-engaging bispecific antibodies (TCBs) as a model therapy. We demonstrate that organoid–immune cell co-cultures are able to capture clinical toxicities overlooked by animal models and shed light on the cellular mechanisms that underpin these effects. Combined efficacy and safety studies in matched healthy and tumour organoids, complemented with multiplexed immunofluorescence (mIF) technology, provided important and clinically relevant insights into the cellular interactions (and differences therein) that mediate epithelial killing during antitumour activity and immune-mediated damage of healthy tissue. Finally, we show that organoids reveal donor-dependent differences in TCB safety liabilities, which could be attributed to variations in target expression among patients. Therefore, we present a versatile and patient-relevant model for the preclinical safety-and-efficacy testing of immunotherapies to help understand and therapeutically exploit fundamental mechanisms of immune-triggered epithelial killing and to guide the optimization of the clinical application of immuno-oncology drugs.

## Results

TCBs are antibodies engineered to recognize two different epitopes: a cancer antigen and, typically, the CD3 T-cell receptor[21,22]. By physically crosslinking the target to the effector cells, the latter are potently activated and instructed to lyse the malignant cells. However, target antigens are rarely restricted to tumours; they are often also expressed in healthy organs, resulting in off-tumour T-cell activation and damage to the target-expressing healthy cells[11]. For example, TCBs targeting the epithelial cell-adhesion molecule (EpCAM) and carcino-embryonic

antigen (CEA), were developed to treat solid tumours, but were found to trigger diarrhoea in phase I clinical trials, suggesting on-target off-tumour intestinal reactivity, consistent with EpCAM and CEA expression in the healthy intestine[23,24]. Endoscopic examination of a patient treated with the EpCAM TCB revealed epithelial cell damage and mononuclear immune-cell infiltration into the mucosa, accompanied by elevated serum levels of inflammatory cytokines IFNγ, IL-6 and IL-8 (ref. 23). Importantly, these effects were not captured by preclinical in vitro and animal models, probably owing to immunological differences between the species and the absence of human-specific isoforms or tissue-relevant expression of targets in non-human primates and mice[25,26]. We sought to examine whether patient-derived intestinal organoids can be used to recapitulate clinical toxicities that were triggered by EpCAM TCB and model toxicities of CEA-targeting TCBs that are under development.

Small intestinal (SI) and colon organoids were established from healthy margins of patient surgical resections[27]. We used immunohistochemistry (IHC) to characterize the SI and colon organoids towards the tissue of origin. By IHC, the presence of both intestinal stem cells and differentiated cell types (enterocytes and goblet cells) was confirmed, the latter increasing from day 3 to day 5 after switching to organoid differentiation conditions (Supplementary Fig. 1a,b). A time-course evaluation revealed the best treatment time to be between day 3 and day 5 of culture, optimizing between cell differentiation stages and viability (Supplementary Fig. 1b). Next, we assessed the expression of TCB target proteins CEA and EpCAM within organoids and compared it to that of parental intestinal samples. Histological IHC evaluation revealed the presence of EpCAM and CEA in both the healthy SI and colon, confirming the availability of the indicated target for TCB binding (Fig. 1a). Both target proteins were likewise expressed in SI and colon organoids at physiologically relevant morphological patterns (Fig. 1b): EpCAM exclusively localized to cellular junctions, whereas CEA showed apical localization, akin to that observed in parental samples. The physiological expression of target antigens within intestinal organoids qualifies these models for the assessment of target-dependent effects associated with TCBs.

Bearing in mind that TCBs exert their effects by simultaneous engagement of epithelial and immune cells, we supplemented the intestinal organoid culture with an immune compartment comprising allogeneic human peripheral blood mononuclear cells (PBMCs) (Fig. 1c). However, instead of culturing immune cells and organoids in suspension or aggregating them at the bottom of wells, as done conventionally[28,29], we co-encapsulated them in solid 3D hydrogels. The solid extracellular matrix (ECM) surrounding the organoids and PBMCs is a more faithful mechanical representation of intestinal tissue and allows simulating crucial immunological processes beyond contact-dependent targeting, including bystander signalling, immune cell migration and infiltration.

Next, we treated the immunocompetent organoid model with EpCAM TCB and monitored immune-mediated epithelial cell lysis by caspase-3/7 induction (Fig. 1d and Supplementary Fig. 1c,d), on the basis of clinical observations of epithelial damage among TCB-treated patients[23]. In addition, we evaluated potential toxicities

---

**Fig. 1 | Patient-derived intestinal organoids co-cultured with PBMCs recapitulate physiological target-expression patterns, enabling the 'back-translation' of TCB-induced on-target off-tumour toxicity. a**, Expression of CEA and EpCAM target antigens in healthy small intestine and colon primary tissue as well as in patient-derived organoids captured by chromogenic DAB (brown) staining at ×20 magnification. Scale bar, 100 μm. **b**, Quantification of the DAB staining by area quantification of individual colon organoids (n > 50). Scale bar, 100 μm. Red line displays mean. **c**, F-actin⁺ outlined organoids (orange) co-cultured with bright DAPI⁺ (blue) PBMCs displayed as maximum intensity projection of a z-stack of ~100 μm. ×20 magnification; scale bar, 200 μm. **c′**, A 3D reconstruction of **c** highlights spatial arrangement of PBMCs around

the organoid (x, y, z axes). **d**, Schematic of low-resolution imaging assay to capture on-target off-tumour toxicity using an organoid–PBMC co-culture. Organoids (5-d expanded and 3-d differentiated) were collected and resuspended with PBMCs before assessing the TCB treatment by brightfield and IF imaging. Schematic created with BioRender.com. **e**, Representative single tiles of merged brightfield and caspase-3/7 IF (green) images of the co-culture treated with EpCAM, CEA(hi), CEA(lo) and non-targeting TCB (0.1–10 μg ml⁻¹) over a time course of 72 h at ×5 magnification. Scale per tile, 500 μm. **f**, Heat map of quantified caspase-3/7 arbitrary fluorescence units (a.f.u.) in >20 segmented organoids per well (n = 3) for each treatment condition (0.1–10 μg ml⁻¹) across time. All displayed experiments in this figure were replicated at least five times, yielding similar results.

of CEA-targeting antibodies, testing a high- and a low-affinity molecule: CEA(hi) TCB[30] and CEA (lo) TCB[31]. An antibody that can bind T cells but not epithelial cells (NT TCB) was used as a control for T-cell activation independent of epithelial binding. The model revealed targeting of organoids by all epithelium-targeted molecules, as evidenced by robust time- and concentration-dependent induction of apoptosis (Fig. 1e,f).

Importantly, the system was sensitive to parameters such as target expression and antibody affinity: in line with the higher EpCAM accessibility (Fig. 1a), EpCAM TCB triggered more rapid and severe organoid cytotoxicity compared with the CEA-targeted molecules. Likewise, CEA(hi) TCB was more damaging than CEA(lo) TCB (Fig. 1e,f). Importantly, these results are in line with clinical reports of more frequent

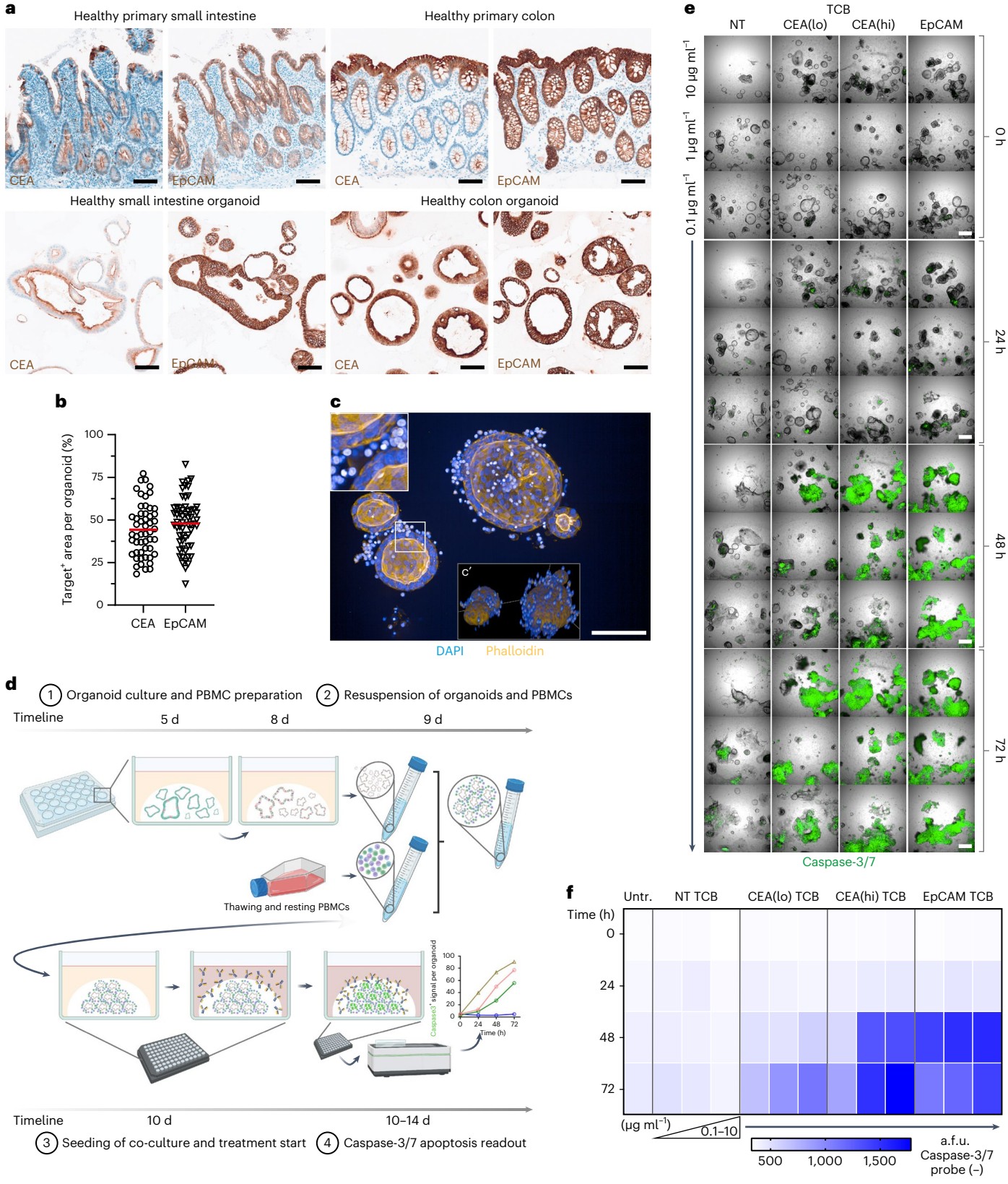

and severe intestinal adverse events associated with EpCAM-targeting bispecifics compared with those binding CEA[23,24]. Together, these data suggest that organoids can provide a robust and sensitive system for modelling on-target TCB-mediated toxicities in healthy organs.

The mechanisms by which TCBs exert their effects in the context of on-target off-tumour toxicity remain incompletely understood. One unanswered question relates to the cellular and molecular drivers of the outcomes, along with their respective temporal dynamics and mutual interactions. An advantage of our model is the possibility to track the evolution of an immune response following TCB administration, within the context of a physiologically relevant ECM-embedded epithelium. To this end, we monitored co-cultures of PBMCs and intestinal organoids, supplemented with TCB molecules (EpCAM, CEA(hi) and CEA(lo), along with the NT TCB control) at three different concentrations (0.1, 1 and 10 µg ml⁻¹). Sacrificial wells from each condition were collected and digested at 5, 24, 48 and 72 h after TCB treatment and phenotypically assessed via flow cytometry to provide a granular characterization of immune cell kinetics in response to the different TCB treatments. Initial analysis of intracellular TNFα, a pivotal pro-inflammatory cytokine responsible for intestinal inflammation and damage[32], demonstrated a detectable response principally localized to a population of GzmB⁺ CD8⁺ T cells and CD45RO⁺ memory CD4⁺ T cells (Supplementary Fig. 2a,b). Therefore, we focused our subsequent phenotypic analysis on these two responder populations. Following antigen engagement, T cells internalize and degrade their surface T-cell receptor (TCR)-CD3 complex in a manner proportional to the magnitude of TCR stimulation[33]. Therefore, we used the intensity of detectable CD3 as a surrogate for ranking the level of stimulation provided by each of the TCBs (Fig. 2a). CD3 downregulation peaked between 24 h and 48 h after TCB administration in both the CD8⁺ and CD4⁺ T-cell fractions, and was accompanied by a temporal pulse induction of TNFα and IFNγ, two fundamental cytokines that synergize to orchestrate potent inflammatory responses (Fig. 2b,c). In contrast, GzmB, a serine protease that mediates cytolytic targeting, was induced in a time-dependent manner, progressively increasing from 24 h until the end of the co-culture (Fig. 2d). These dynamics suggested that GzmB induction was a secondary response to the initial inflammatory milieu created by TCB-induced T-cell activation. At 72 h after TCB administration, T cells transitioned into a growth phase, as demonstrated by detection of intranuclear Ki67 (Fig. 2e).

Overall, the magnitude of the response correlated with the degree of expected target antigen binding. Namely, TCBs targeting EpCAM, which is more accessible than CEA, triggered the strongest pattern of TCR downregulation, cytokine/GzmB expression and cell cycling (Fig. 2a–f, red squares). Whereas CEA(hi) also induced a robust response, CD3 expression after CEA(lo) treatment remained constant, suggesting only modest CD3 engagement triggered by this molecule (Fig. 2a–f, purple triangles). The dose titration of each TCB showed that, unsurprisingly, higher TCB concentrations induced stronger T-cell activation (Fig. 2a–e). Most intriguing was GzmB expression across the different TCB concentrations. EpCAM TCB, due to the accessibility of the EpCAM antigen, invoked a potent induction, regardless of the concentration of TCB tested here. CEA(hi) TCB, however, transitioned between complete absence of induction at the lowest concentration

(0.1 µg ml⁻¹) to near complete response at the middle concentration (1 µg ml⁻¹), particularly within the responding CD4⁺ fraction (Fig. 2f). The magnitude and kinetics of GzmB induction are concordant with the level of organoid targeting and expression of caspase-3/7 within the epithelium for each treatment (Fig. 1e,f), confirming this enzyme as a key driver of the cytotoxicity. Soluble cytokine analysis from the supernatants at each timepoint demonstrated a similar transition between complete absence of inflammatory cytokines at the lowest concentration of CEA(hi) TCB to near complete response at the middle concentration. IFNγ, IL-2, IL-4, TNFα and GM-CSF all followed this pattern of expression (Fig. 2g). Likewise, cytokines secreted in response to CEA(lo) TCB treatment were nearly undetectable, whereas EpCAM TCB induced high levels of cytokines at all three concentrations, in line with the apoptosis outcome associated with each molecule (Fig. 1e,f). It is important to highlight that the induction of IFNγ, IL-6 and IL-8 in response to EpCAM TCB treatment is consistent with elevated serum levels of these cytokines in patients treated with the same molecule[23]. Collectively, these data depict the kinetics of TCB-induced intestinal targeting, aligning with reported patient safety profiles from clinical trial data and demonstrating the utility of our model, with its sensitivity to TCB dose, antigen affinity and target accessibility.

Although flow cytometry-based phenotyping provided valuable insight into the immunological mechanisms that drive TCB-associated intestinal toxicities, it reveals only a partial picture. Primarily, it does not provide information on the spatial interactions within and between the epithelial and immune compartments that underlie the outcomes. For example, it is unclear whether cytotoxic effects are primarily and exclusively driven by T cells that are incidentally located in the vicinity of tumour cells and target, or, if following initial lysis events and concomitant inflammation, distant cells are recruited and actively migrate to the site to inflict further damage. To analyse the TCB-mediated immune activation in a spatiotemporally resolved manner, we developed a 7-plex mIF approach to visualize pan-cytokeratin⁺ epithelial cells (panCK), induction of apoptosis (caspase-3/7), CD4⁺/CD8⁺ T cells, CD20⁺ B cells and CD14⁺ monocytes, as well as to visualize and quantify target expression (Supplementary Fig. 3a). Importantly, both preservation of spatial information and histological sectioning were possible owing to the 3D format of the system, wherein organoids and immune cells are housed within solid ECM, rather than in suspension or aggregation.

In line with results derived from live imaging (Fig. 1), we observed an increase in caspase signal within the organoids following TCB treatment, which began appearing around the 24 h timepoint and culminated at 72 h, coupled with necrosis of the epithelial layer (Fig. 3a). The mIF approach also unveiled the exquisite and, in some cases, unexpected organoid–immune cell interactions that accompany and probably drive the outcome. Whereas CD4⁺ and CD8⁺ T lymphocytes in the control condition appear to be small, round and stationary throughout the experiment, TCB-stimulated cells became larger, oval and underwent extensive migration towards the epithelium, resulting in organoids that were fully immune-infiltrated by the end of the experiment (Fig. 3a). Further, we observed T-lymphocyte infiltration between cells of the organoid epithelium, which resembles the integration of intra-epithelial lymphocytes within the intestinal barrier[34] (Fig. 3a,

**Fig. 2 | TCB-triggered immunological activation cascades of CD45RO⁺ CD4⁺ and GzmB⁺ CD8⁺ immune subsets yield mechanistic insight into the TCB mode of action.** The figure follows the CD45RO⁺ CD4⁺ and GzmB⁺ CD8⁺ T-cell fractions over the treatment period (hereafter abbreviated as 'both subsets'). The figure displays data for both subsets, following the immunological cascade and cytokine release at 10 µg ml⁻¹ across the entire treatment duration (left panels) and at three different concentrations (0.1–10 µg ml⁻¹) (right panels) of the indicated TCB at specific timepoints (denoted by asterisks). **a**, TCR (CD3) MFI highlights TCR internalization upon TCB stimulus in both subsets. **b–d**, Brefeldin A and Monensin were used to trap cytokines secreted in individual cells. Here we detected pro-inflammatory cytokine release by means of TNFα (**b**) and IFNγ (**c**) in both subsets, followed by GzmB exocytosis (**d**). **e**, Proliferative state of CD45RO⁺ CD4⁺ and Ki67⁺ GzmB⁺ CD8⁺ by Ki67⁺ antibody staining reiterates potent activation of immune cells. **f**, Concatenated contour plot of GzmB⁺ intracellular expression in both T-cell subsets 48 h post treatment, in which each columnar cloud represents the individual condition indicated. **a–f**, $n = 2$. **g**, Heat map of multiplex cytokine analysis performed on supernatants from treated wells across all timepoints and TCBs administered. Normalized per cytokine, mean per condition plotted ($n = 3$).

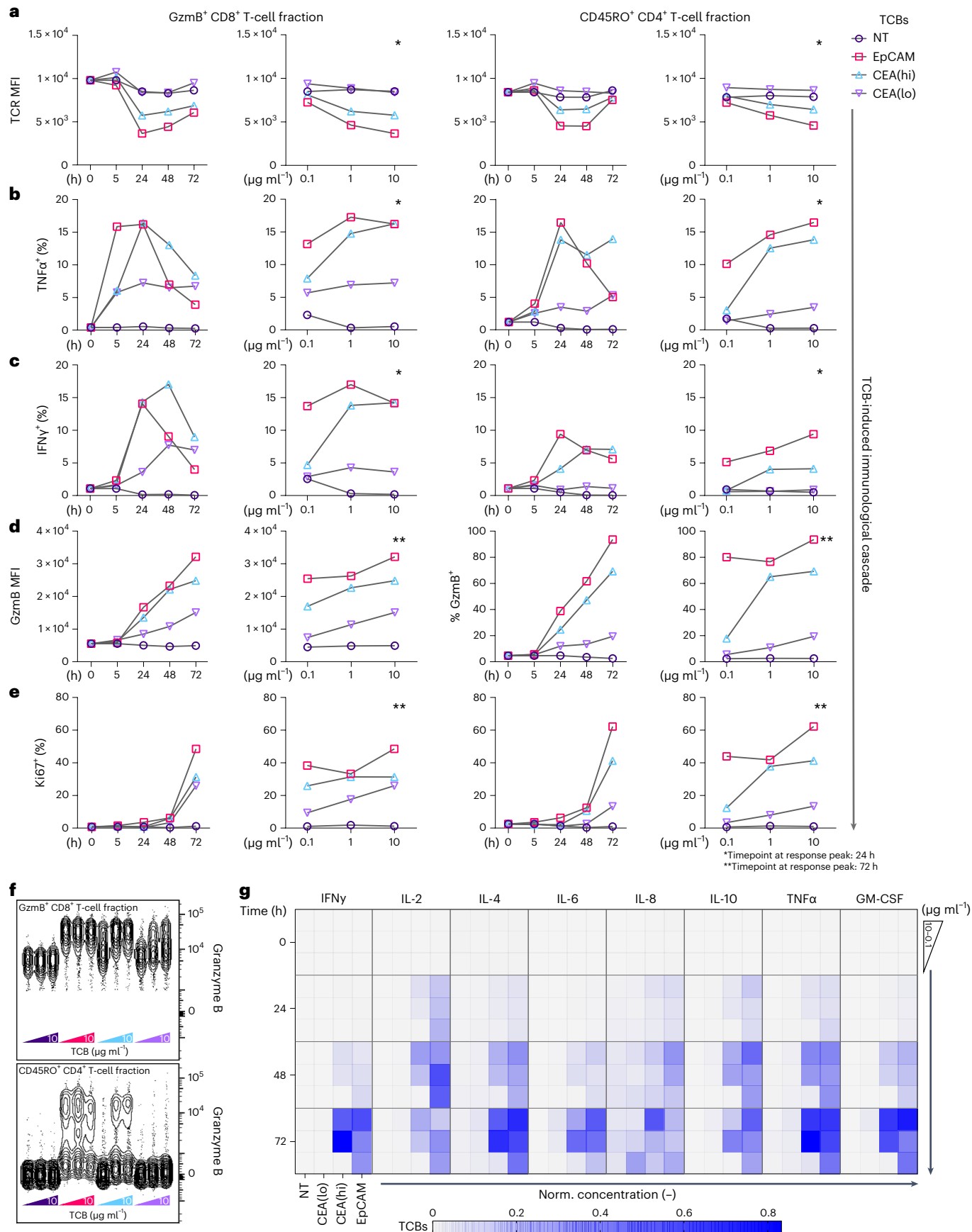

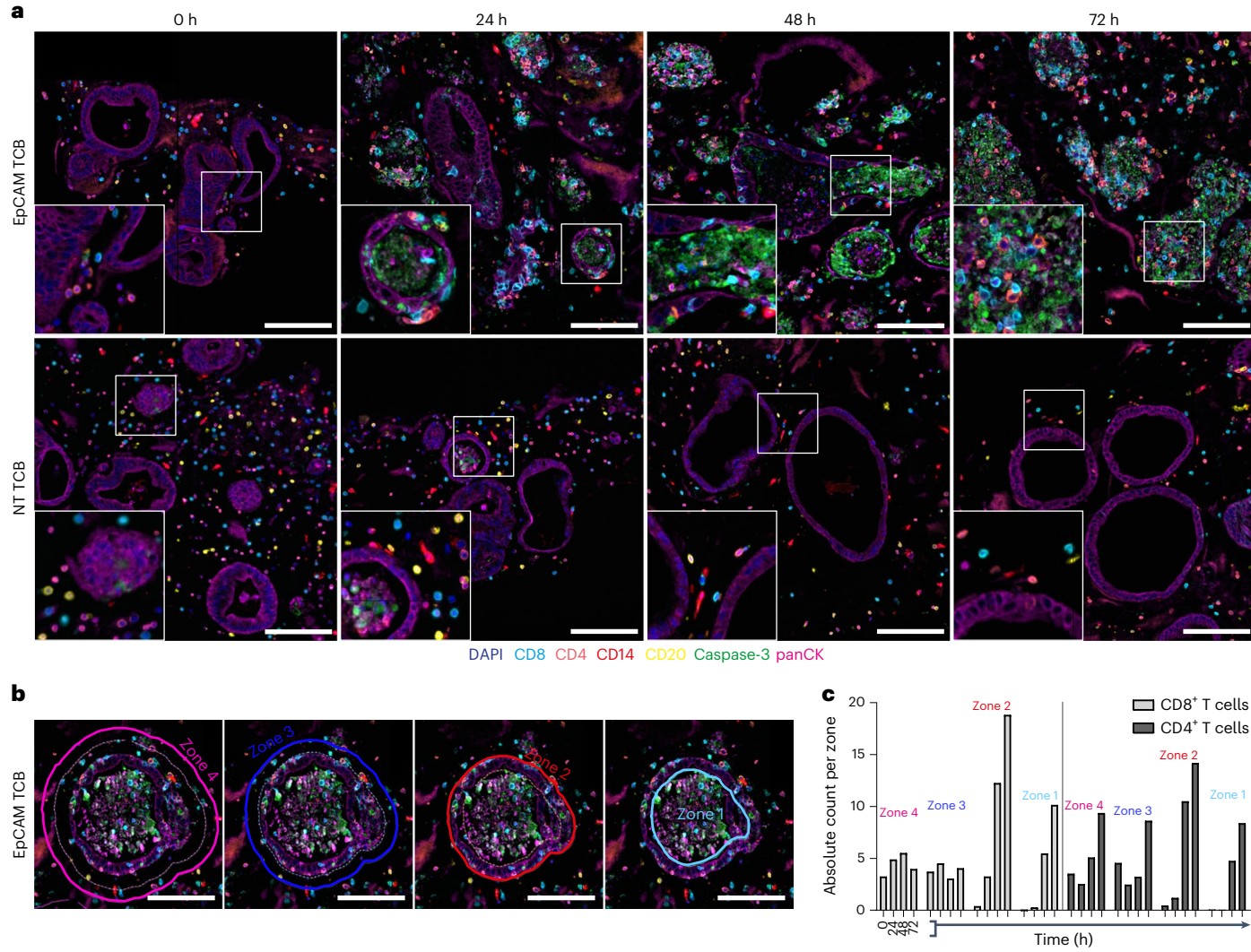

DAPI CD8 CD4 CD14 CD20 Caspase-3 panCK

**Fig. 3 | mIF-based dissection of immune–epithelial interactions during TCB-mediated toxicity in intestinal organoids. a**, Representative single tiles (×20 magnification) of the 7-plex mIF images across all EpCAM TCB and NT TCB treatments (10 μg ml⁻¹) across 0–72 h time course. PanCK⁺ organoids are surrounded by CD4⁺ (orange), CD8⁺ (turquoise), CD14⁺ (red) and CD20⁺ (yellow) immune cells. Caspase-3 (green) captures TCB-mediated off-tumour toxicity in the organoids. Nuclei are stained with DAPI (blue). Scale bar, 100 μm. **b**, An image at 24 h post EpCAM TCB administration highlights concentric partitioning

in 25 μm margins around the initial ROI of the colon organoid (zone 2, red solid line): one inward margin (zone 1), two outward margins (zones 3 and 4). Solid line, inclusion; dotted line, exclusion. Scale bar, 100 μm. **c**, Mean of the absolute counts of CD4⁺ and CD8⁺ T lymphocytes in the zones around individual organoids across time highlights infiltration upon EpCAM TCB (10 μg ml⁻¹) application. All displayed experiments in this figure were replicated at least three times, yielding similar results.

magnified inserts). This process appears to precede the induction of apoptosis and may therefore be an essential step of the interaction cascade that ultimately leads to organoid demise. These observations are also in line with reports of intestinal immune cell infiltrates in patients experiencing severe diarrhoea upon TCB treatment[23].

To quantify these findings, we devised an image analysis strategy to assess the spatial distribution of immune cells in relation to organoids over time. Briefly, by training a deep-learning algorithm (DenseNet AI 2, HALO AI v.3.4.2986.209) to distinguish between matrix, organoid and immune cells, we created individual organoid ROIs and generated inward and outward concentric partitions to denote the interior of the organoids (zone 1), the organoid epithelial barrier (zone 2) and the proximal and distal extracellular space (zones 3 and 4) (Fig. 3b and Supplementary Fig. 3a,b). Quantification of the number of T cells within these regions over time following EpCAM TCB administration confirmed the visual observations (Fig. 3c and Supplementary Fig. 3f): the numbers of both CD4⁺ and CD8⁺ T cells within zones 1

and 2 substantially increased over time (Supplementary Fig. 3i). Furthermore, unlike the low-resolution live imaging, which provided an arbitrary and relative measure of damage, the high-resolution quantification of the caspase signal shows the physical extent of organoid damage over time (Supplementary Fig. 3g,h).

To support the translation of therapeutic molecules into the clinic, safety data are meaningful when presented in conjunction with efficacy data, that is, compared against antitumour activity of the drugs in the form of a safety margin. Therefore, we extended our system to simultaneously assess both killing of colorectal cancer organoids (hereafter referred to as tumouroids) and intestinal toxicity using donor-matched healthy organoids (Fig. 4). Before performing functional studies, we assessed the cellular architecture and expression of the targets introduced above within organoids and tumouroids using haematoxylin and eosin (H&E) and IHC staining (Fig. 4a). In both cases, organoids recapitulate the architecture of the respective native tissue (Fig. 1a): whereas healthy organoids feature a simple epithelial monolayer, tumouroids

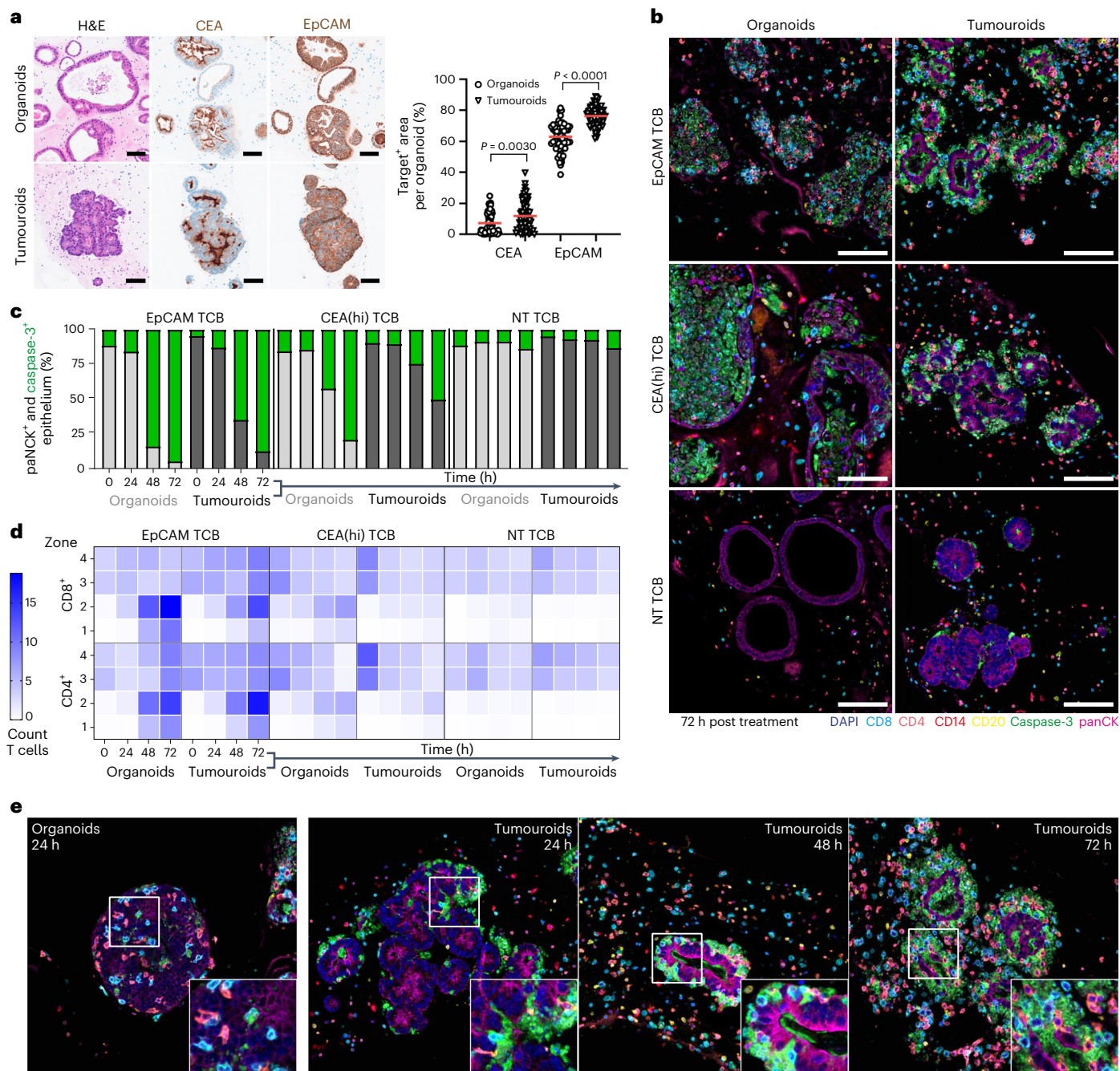

**Fig. 4 | Joint efficacy–safety analysis of TCB effects reveals differences in immune-cell engagement with healthy and tumour organoids. a**, Left: comparison of cellular architecture and target antigen expression between organoids and tumouroids by H&E and IHC at ×20. Right: quantification of target antigen expression by positive area for each antigen per individual organoid ($n_{OrgCEA}$ = 68, $n_{TumCEA}$ = 76; $n_{OrgEpCAM}$ = 60, $n_{TumEpCAM}$ = 85). ROUT outlier analysis was performed with a Q coefficient of 2%. Statistical analysis was performed using one-way ANOVA. Red line displays mean. Scale for both H&E and IHC, 100 μm. **b**, Representative single tiles (×20 magnification) of the 7-plex mIF images across all TCB treatments (10 μg ml⁻¹) at 72 h, displaying panCK⁺ organoids and tumouroids (magenta) surrounded by CD4⁺ (orange), CD8⁺ (turquoise), CD14⁺ (red) and CD20⁺ (yellow) immune cells. Caspase-3 (green) captures TCB-triggered immune-induced apoptosis, nuclei are stained with DAPI (blue). Scale bar, 100 μm.

**c**, Sum of panCK⁺ (grey) and caspase-3⁺ (green) epithelium of the organoids and tumouroids, respectively, detected in zone 2 (on epithelium) across the different TCB treatments and time ($n$ = 3). **d**, Quantification of the 7-plex mIF images represented in **b** and **e**. Heat map of the absolute counts of T-cell subsets within the different zones of individual organoids and tumouroids across time and TCB treatment. **e**, Single tiles of the 7-plex mIF staining at ×20 magnification (colours explained in **b**) highlights substantial immune-intercalation dissimilarities between healthy and cancerous epithelium treated with EpCAM TCB (10 μg ml⁻¹). Organoid image: highly CD4⁺ and CD8⁺ T-cell infiltrated organoids 24 h post administration. Tumouroid images: progressively killed tumouroid (caspase-3⁺ apoptotic bodies) over time, but devoid of T lymphocytes within the inner core of the tumouroid. Scale bar, 100 μm. All displayed experiments in this figure were replicated at least three times, yielding similar results.

are heterogeneous and densely packed, nearly lacking a luminal space (Fig. 4a). Furthermore, unlike organoids, which contain the full diversity of differentiated intestinal cells, tumouroids appear to mainly comprise progenitor and secretory cells (Supplementary Fig. 4a). As expected, tumouroids featured higher target levels (EpCAM and CEA) compared with organoids, although the differences were modest (Fig. 4a).

We next applied the mIF method introduced above to analyse epithelial damage, immune cell behaviour and activation within TCB-treated organoids and tumouroids (Fig. 4b and Supplementary Fig. 4b). Although both tissue types were targeted by lymphocytes, immune infiltration and epithelial cytolysis appeared to be more extensive in healthy organoid co-cultures despite slightly higher target expression within tumouroids. In particular, TCB-triggered apoptosis within tumouroids appeared to be delayed compared with that within organoids, as indicated by the higher proportion of cytokeratin-positive living epithelial cells that persist at the later timepoints (48 and 72 h; Fig. 4b,c). Indeed, whereas organoids were thoroughly destroyed at 72 h after treatment, tumouroids preserved an intact core (Fig. 4b). To shed light on the cellular mechanisms that underlie this finding and bearing in mind that epithelial targeting is mediated by immune cells, we quantified the spatiotemporal dynamics of T lymphocytes in the various organoid zones, as outline above, during the process (Fig. 4d). Comparing organoids and tumouroids, we observed notable differences in their susceptibilities to T-cell infiltration. Both CD4+ and CD8+ lymphocytes appeared to integrate faster and more extensively within the healthy organoid epithelium than within the tumouroids. The results obtained from cell number quantification were confirmed by high-magnification images of stained sections: whereas TCB-treated organoids featured a high number of T cells inserted between epithelial cells within 24 h, T cells within tumouroid samples appeared to be largely sequestered to the basal side of the epithelium and rarely integrated even 72 h post TCB application (72 h; Fig. 4e). To investigate whether our observations can be extended to other donors, we generated two additional organoid–tumouroid pairs and repeated the experiments. Consistently, tumouroids experienced lower extents of damage and T-cell infiltration compared with organoids (Extended Data Fig. 1c–f). Comparing outcomes of experiments performed with allogeneic vs autologous PBMCs, we obtained similar results in the case of healthy organoids, that is, allogeneic and autologous PBMCs induced comparable levels of apoptosis upon TCB treatment. In contrast, we observed that allogeneic PBMCs targeted tumouroids more extensively than autologous ones. It is possible that any neoantigen-driven (bystander) T-cell activation is further amplified by an alloreactive response.

Higher efficiency of immune cell integration within the epithelium provides a compelling explanation for the increased TCB-mediated killing of healthy organoids, despite expressing lower target levels than tumouroids. Intra-epithelial lymphocytes and activated peripheral T cells conventionally infiltrate within intestinal epithelium by directly binding E-cadherin via the integrin αE(CD103)β7, itself upregulated during inflammation[34,35]. Intriguingly, we observed attenuated E-cadherin expression within tumouroids compared with organoids (Extended Data Fig. 2a,b). To explore whether TCB-activated T cells infiltrate within organoids by binding to E-cadherin and whether T-cell exclusion within tumour organoids could be explained by its down-regulation, we used an αE(CD103)β7-inhibitor cocktail comprising a recombinant human monoclonal antibody etrolizumab[36] and an anti-integrin β7 monoclonal antibody[37] to inhibit the CD103-E-cadherin interaction within TCB-treated co-cultures (Extended Data Fig. 2c). To our surprise, neither organoid apoptosis nor T-cell infiltration were affected (Extended Data Fig. 2d,e). IHC and flow cytometry analysis for CD103 expression within TCB-treated co-cultures revealed no upregulation of this protein, suggesting that T cells use a different mechanism to integrate within organoids (Extended Data Fig. 2f,g). Bearing in mind that TCB targets are surface molecules, we believe that in the context of the toxicities described here, lymphocyte epithelial infiltration is driven by CD3–target (CEA/EpCAM) interaction.

Given the substantial differences in epithelial architecture between the two tissue types (Fig. 4a and Extended Data Fig. 2a,b), we considered organoid/tumouroid morphology as a potential factor underlying the differential infiltration and damage. To test our hypothesis, we established additional tumour lines of varying morphologies, including those that resembled the simple monolayer and large lumens of healthy organoids and others that were more densely packed and contained smaller lumens (Extended Data Fig. 3a). The co-culture experiments revealed no correlation between morphology and tumouroid apoptosis or lymphocyte infiltration; that is, the tumouroid line featuring a simple monolayer was not targeted most extensively, as we had expected (Extended Data Fig. 3b–e). The differences in apoptosis and T-cell infiltration most strongly correlated with target (EpCAM) expression (Extended Data Fig. 3c,f,g), suggesting that target expression is the most robust determinant of organoid susceptibility to TCB-mediated damage, rather than morphology. We still do not understand what drives the differences in infiltration between tumouroids and organoids of the same donor. This question warrants a separate in-depth study that may reveal interesting fundamental differences between tumour vs healthy epithelium that may be leveraged therapeutically.

We next considered the utility of our system in precision medicine applications, by exploring whether it can capture patient variations in susceptibility to cancer-immunotherapy responses, in this case toxicity. To this end, we generated 14 additional SI and colon organoid lines using biopsies derived from different patients. The organoids were then treated with EpCAM- and CEA-targeting TCBs in co-culture with PBMCs. Quantification of TCB-mediated organoid killing revealed substantial differences in response within the organoid cohort (Fig. 5a,b), wherein certain organoid lines were targeted readily, while others seemed refractory to damage. As expected, EpCAM TCB and CEA(hi) TCB led to higher overall targeting compared with the low-affinity CEA-binding TCB.

Intriguingly, the effects of both EpCAM and CEA(hi) TCB were highest in a shared subset of organoid lines, suggesting a common

**Fig. 5 | The organoid model detects donor-dependent differences in TCB-triggered toxicity. a**, Representative single tiles of merged brightfield and IF images of the 14-organoid donor lines co-cultured with PBMCs treated with EpCAM, CEA(hi), CEA(lo) and NT TCB (0.5 μg ml⁻¹) at 48 h post TCB administration. ×5 magnification; scale bar per tile, 1 mm. Organoid donor number is random, following no particular order. **b**, Heat map of quantified caspase-3/7 fluorescence signal in dozens of segmented organoids per well (n = 3) for each TCB treatment across time. Mean fluorescence signal for each TCB condition normalized to the mean a.f.u. detected in the NT TCB control at each timepoint. The 14 different patient-derived organoid lines are displayed on the x axis and ordered by extent of apoptosis experienced. **c**, Representative IHC of CEA (top) and EpCAM (bottom) expression of the 14 embedded organoid lines at ×40 magnification. Organoid donors ordered by expression levels. Scale bar, 50 μm. **d**, Quantified target expression levels of cross sections in **c** for CEA (top) and EpCAM (bottom) indicated as positive area for each individual organoid. Donors are ordered by increasing target expression. **e**, Data in **d** as boxplots, with whiskers showing all points (minimum to maximum) of the mean target expression across all donors distinguished between the indicated intestinal regions for each organoid line (n = 7). Unpaired t-test (two-tailed) was performed. **f**, Correlation plots of the target expression between CEA or EpCAM and normalized caspase-3/7 signal of the respective TCB. R² is provided per plot. **g**, Data as boxplots, with whiskers showing all points (min. to max.) of the mean of normalized caspase-3/7 across all donors distinguished between the indicated intestinal regions for each organoid line at 48 h post administration (n = 7). Unpaired t-test (two-tailed) was performed. All displayed experiments in this figure were replicated at least three times, yielding similar results.

mechanism of susceptibility to immune-mediated damage for these patients. Bearing in mind that target abundance is one of the primary factors that govern the potency of TCB effects, we considered patient-specific variations in target expression as the factor underlying the functional outcome. We performed IHC analysis of both EpCAM and CEA expression across all 14 organoid lines (Fig. 5c). Visual inspection

and signal quantification revealed notable variations in CEA expression across donors and between organoids derived from different intestinal segments, with colon organoids expressing much higher levels compared with SI organoids (Fig. 5d,e). In contrast, EpCAM expression was less variable between different organoid lines and intestinal regions (Fig. 5c–e). Importantly, these data are consistent with the

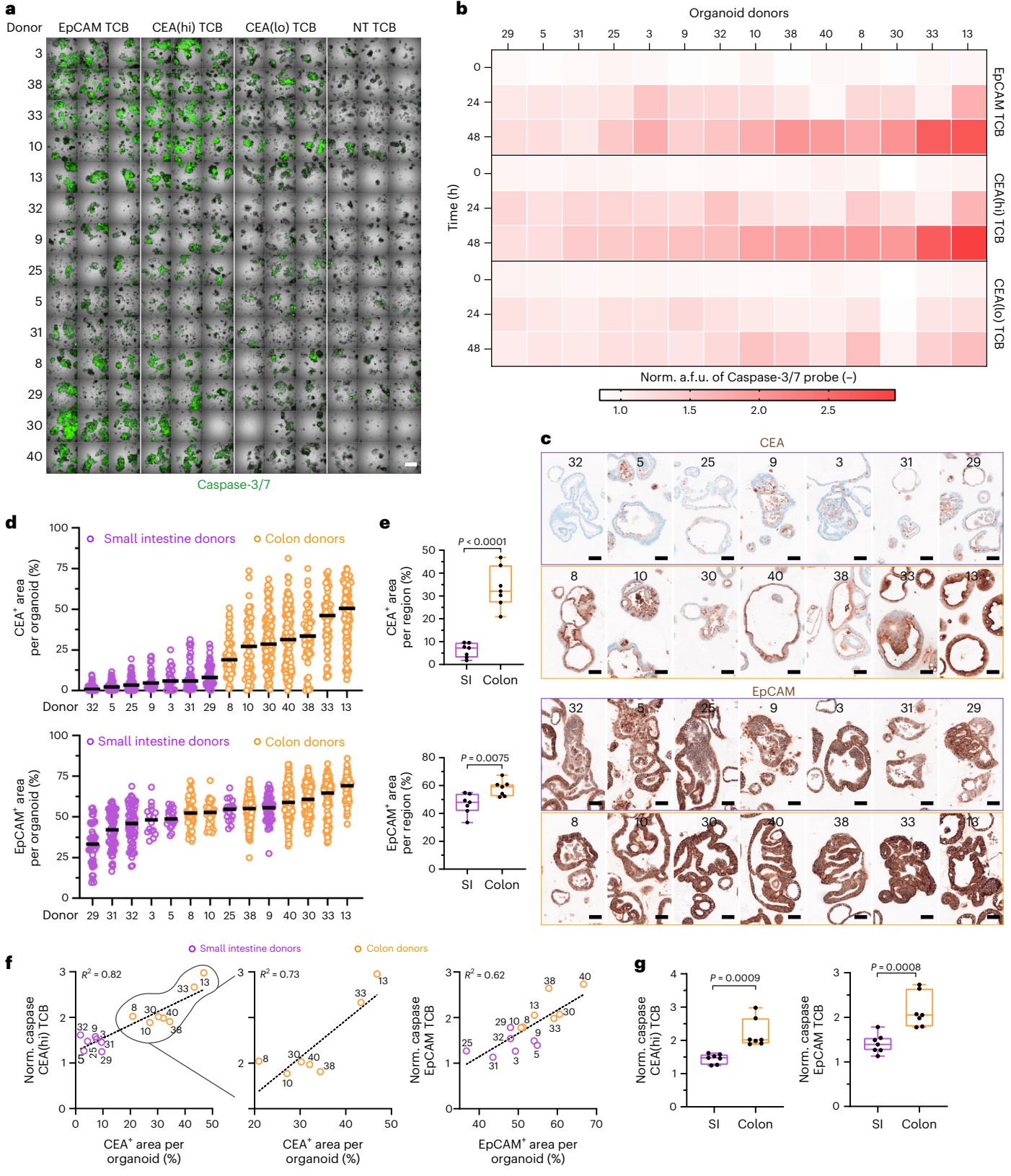

expression of these proteins in parental samples: CEA expression is higher in the colon, whereas EpCAM is expressed at comparable levels between SI and colon (Fig. 1a and Human Protein Atlas), suggesting that the organoid system faithfully captures regional variations of protein expression within the native intestine. We noted a positive correlation between toxicity outcome and target abundance for both the EpCAM- and CEA-binding molecules (Fig. 5f and Supplementary Fig. 5), implicating, as expected, target expression as a major determinant and potential predictor of adverse events. However, it is important to highlight meaningful outliers such as donor 8, which, despite featuring the lowest expression of CEA among the colon-derived lines, was subject to high immune-mediated targeting. This result implies that target expression alone, despite overall positive correlation, is not the sole factor that governs organoid killing. Further, we observed clear separation of CEA TCB-induced organoid killing based on intestinal region, which is expected, bearing in mind the substantial enrichment of CEA in the colon (Fig. 5e–g). However, our data likewise show a significantly lower response of SI organoids to EpCAM TCB despite high levels of EpCAM expression, suggesting that small intestinal epithelium may inherently be more refractory to immune-mediated injury. Bearing in mind that these experiments were conducted using allogeneic PBMCs, we found it important to also test how PBMCs from different donors influence the outcome. We repeated the co-culture experiments with a subset of 6 organoid lines from the original 14, with low-CEA- and high-CEA-expressing donors being represented. Each organoid line was co-cultured with PBMCs derived from three individuals (Supplementary Fig. 6a). Consistently, donors expressing higher CEA levels were targeted more extensively, and we derived a strong positive correlation between organoid apoptosis and target expression (Supplementary Fig. 6b–e). The results were reproducible across the three PBMC donors (Supplementary Fig. 6b,d). We also explored how the extent of T-cell infiltration and activation varies in co-cultures containing organoids from different individuals, using the mIF approach and multiplexed ELISA, respectively. Organoid lines that were heavily damaged were also most extensively infiltrated by both CD4+ and CD8+ T cells (Supplementary Fig. 6e,f). In addition, we detected the highest induction of pro-inflammatory cytokines in lines 8, 9 and 13, which respectively express high levels of target, undergo most extensive damage and are most infiltrated by immune cells (Supplementary Fig. 6g).

Thus, it appears that target abundance, intestinal region and additional donor-specific factors all contribute towards determining the functional effects of TCB treatment on intestinal organoids. It would be interesting to further dissect the mechanisms underlying these differences in response and potentially leverage them as markers to predict or a means to minimize immune-related toxicities associated with cancer-immunotherapy drugs.

## Discussion

The ultimate goal of immuno-oncology is to develop treatment strategies with a favourable therapeutic window, that is, therapies that preferentially target tumour cells without collateral damage to healthy tissues. Owing to factors including poorly understood tumour immunology and microenvironment, scarcity of tumour-restricted antigens, inadequate preclinical models and unguided clinical application, cancer-immunotherapy approaches have yet to realize their full potential, especially in treating solid tumours. Here we have introduced an organoid-based approach that can expand the therapeutic power of cancer-immunotherapy drugs by (1) allowing more accurate preclinical assessment of efficacy and safety profiles, (2) shedding light on fundamental cellular and molecular mechanisms that govern immune-mediated killing of tumour and healthy cells and (3) enabling precision clinical application by guiding patient selection.

The utility of organoid technology as a preclinical system for cancer-immunotherapy drug efficacy testing or studying immune effector functions against cancer cells has already been demonstrated[20,28,38–40]. However, antitumour effects in the clinic can rarely be uncoupled from adverse reactions in healthy tissues, stemming from mis- or undirected immune cell activity. These toxicities severely limit, and in some cases, preclude the clinical application of cancer immunotherapy[9–11]. We demonstrate that healthy patient-derived organoids supplemented with PBMCs are effective in capturing immune-related intestinal toxicities of cancer-immunotherapy drugs, such as TCBs. In particular, the model recapitulated clinical toxicities of EpCAM- and CEA-targeted therapies[23,24] in the form of extensive organoid apoptosis triggered upon treatment. Importantly, mouse models employed during preclinical development failed to predict intestinal risks of EpCAM/CEA-targeting TCBs and CEA CAR T cells[25,26]. We also show that the organoid system can predict toxicity of drug candidates at the preclinical stage and guide their optimization towards more favourable safety profiles. Whereas a first-generation molecule potently binding CEA was found to trigger extensive organoid killing (suggesting high risk of intestinal damage in patients), lowering its binding affinity reduced the effect (Fig. 1f). Lowering TCB target affinity and potency would expectedly lead to a decrease in tumour targeting. However, this is still a viable strategy in cases where toxicities initially prevented escalation to therapeutic doses[23], provided that a modified molecule preserves antitumour efficacy while ensuring safety[31,41].

We should highlight that the approach we describe here is probably not suitable for capturing toxicities associated with immune checkpoint inhibitors (ICIs)[42] whose mechanisms are more complex and, aside from direct T-cell activation, may include cross-reactivity of tumour-specific lymphocytes and B-cell-mediated auto-antibody production. Based on recent studies linking ICI-induced colitis to tissue-resident T cells[6], incorporating these populations within the co-culture assays could be a step towards recapitulating ICI adverse events.

Beyond assessing the gross phenotypic outcome via quantification of apoptosis, we dissected the molecular mechanisms whereby cancer-immunotherapy-stimulated immune cells damage intestinal epithelium. Close correlation between the magnitude and kinetics of GzmB induction in T cells and apoptosis in epithelial cells suggested that this protease is responsible for immune-mediated organoid killing (Fig. 2). However, we also detected an early release of pro-inflammatory cytokines TNFα and IFNγ, both of which are known for their deleterious effects on the intestinal epithelium[32,43]. It would be important to define the respective roles of these species in the safety vs efficacy context, potentially identifying approaches to mitigate effects on healthy tissue without compromising antitumour activity. Indeed, prophylactic TNFα blockade has been shown to uncouple efficacy from colitis-like toxicities during checkpoint inhibitor treatments of xenografted mice[44]. However, a recent study did observe a partial attenuation of TCB antitumour activity upon TNFα blockade, suggesting that alternative mitigation strategies should also be considered[45].

Although disruptive techniques such as flow cytometry provide invaluable molecular insights, they fail to convey information on the spatial interactions between immune cells and organoids, which have not been previously visualized at the single-cell level in either the tumour or healthy tissue context. We developed an histological embedding protocol preserving the locations and interactions of organoids and PBMCs upon fixation. Spatiotemporally resolved quantitative analysis of these sections associated TCB-mediated organoid damage with T-cell proliferation, migration and epithelial infiltration. Among these, particularly surprising was the intra-epithelial insertion of blood-derived T cells, which has not been described before in human organoids in general and appears to be required for efficient epithelial cell killing (Figs. 3 and 4).

Insights into the cellular mechanisms of TCB toxicity in the clinic are difficult to glean because healthy tissues are simply not biopsied during the course of the treatment, which is when the adverse effects are at their most acute. However, one patient treated with EpCAM TCB underwent

endoscopic evaluation 7 d after treatment termination, owing to persistent abdominal pain[23]. The histological findings are similar to the observations we derived from the first organoid model. Damage to the crypt and villus structure of the patient duodenum was observed, which is consistent with epithelial apoptosis. Extensive lymphocyte infiltration into the tissue, including between epithelial cells, was also reported and was recapitulated in our model (Figs. 3 and 4). Measurement of serum cytokines in the clinic revealed increase in the levels of IFNγ, IL-6 and IL-8, all of which were likewise induced in TCB-treated organoid–PBMC co-cultures (Fig. 2g). Given the concordance of selected clinical and in vitro outcomes, we believe that this system can help better understand and identify ways to mitigate clinical toxicities.

Although patient-derived organoids have been pledged to applications in precision oncology[46,47], organoid-based studies capturing variability in patient response have been limited to testing chemotherapy and targeted therapy drugs[48–50]. Using organoids derived from 14 donors and co-cultured with immune cells, we show that the model can reveal inter-donor variability in their susceptibility to TCB-induced damage, opening avenues in precision immuno-oncology. Dissecting the factors that govern the variation in response, we found unsurprisingly that the likelihood for TCB damage of a particular donor correlated with target abundance within the organoid (Fig. 5). Variability in target expression between colon and SI organoids likewise predicted that these regions would experience different risks of on-target toxicities. Unfortunately, current clinical reports of TCB-induced gastrointestinal toxicities are not able to definitively link the effects to a specific region[23,24]. In addition to identifying features that are predictive of patient response, we view this model as a potential companion diagnostic in the clinic. The efficiency and timescales required to generate organoids from patient biopsies, however, are probably not conducive to this application at present. Nonetheless, innovative approaches, including those based on droplet microfluidics[18], could expedite and increase the efficiency of organoid formation, rendering them compatible with clinical application.

We also used the system to explore efficacy and safety effects by comparing responses of transformed and healthy epithelia to TCB treatment. We should underscore that we recorded some differences in the extent of damage inflicted to healthy vs tumour organoids, depending on whether autologous or allogeneic PBMCs were used (Supplementary Fig. 5d). Specifically, we observe enhanced killing of tumour organoids when allogeneic PBMCs are used, which may be the consequence of additional immune cell activation owing to alloreactivity. However, we would be cautious to draw this conclusion based on one experiment. The difference could also be the result of the typically lower viability and quality of autologous PBMCs collected from cancer patients compared with commercially sourced cells. Collectively, all experiments yielded equivalent results regarding questions on the main TCB effects: (1) whether healthy organoids are targeted and (2) whether tumour organoids are targeted more or less in comparison; that is, whether a therapeutic window exists. Therefore, our assay can be reliably used to answer these questions with allogeneic PBMCs, even though it might not be able to provide an exact therapeutic window and absolute TCB concentrations to be used in the clinic.

In all co-culture experiments and across multiple donor pairs, we observed remarkable differences in the ability of T cells to integrate within healthy vs tumour organoids, the latter being more impenetrable and hence more unassailable, despite expressing higher target levels (Fig. 4). Tumour-mediated immunosuppression comes in many forms[51]. Structural and architectural features of the tumour epithelium itself may be an additional underexplored means to evade immune attack. Exploring the mechanisms whereby tumour epithelium shields itself against immune penetration and, conversely, whereby healthy epithelium permits integration, could provide new actionable targets for both enhancing antitumour efficacy and reducing the toxicities of T cell-based immunotherapies.

## Methods

### Human tissue samples
Tissue samples and annotated data were obtained, and experimental procedures were performed within the framework of the non-profit foundation HTCR, including informed patient consent.

### Intestinal organoid and tumouroid cell culture
Supplementary Table 1 provides anonymized basic donor information. After isolation of intestinal crypts following a previously described protocol[27], crypts were embedded in 25 μl droplets of growth factor-reduced Matrigel (356231, Corning) and cultured in a 50% (v/v) mix of Human IntestiCult OGM human basal medium (OGM; 100-0190, StemCell) and organoid supplement (100-019, StemCell Technologies) in a 24-well clear TC-treated plate (3524, Corning Costar). Y27632 (10 μM l⁻¹; 72302, StemCell Technologies) and 1% penicillin/streptomycin (15140-122, Gibco) were spiked into the media after seeding the organoids. Throughout the week of expansion, OGM was changed every 2 d, deprived of Y27632. Organoids were passaged every 7 d, up to a maximum of 25 passages. After 2–3 passages post-isolation, organoid cultures were amenable to experiments. Isolation and passage protocol in ref. 27 were slightly adjusted for our purposes[34]. Briefly, we added 500 μl of Gentle Cell Dissociation reagent (07174, StemCell) before disrupting the domes. After 15 min of incubation at r.t., the suspension was centrifuged at 250 g for 5 min before washing with 'organoid wash' (DMEM-F12 + Glutamax (61870-010, Gibco), 10 mM HEPES buffer (15630-056, Gibco), 1% penicillin/streptomycin, 1% bovine serum albumin (A9647-100G, SIGMA)). Centrifugation was repeated and the supernatant discarded; the pellet was then resuspended and seeded in a Matrigel dome.

Colorectal cancer organoids (termed tumouroids) were obtained and handled as described above, except for media applied. Tumouroids were grown in a 50% (v/v) mix of IntestiCult OGM human basal medium (100-0190, StemCell) and DMEM-F12 + Glutamax, supplemented with 1% penicillin/streptomycin and 10 μM l⁻¹ Y27632. Media change and passages were performed as described for the healthy organoids.

For differentiation of the organoids, OGM was removed after 5 d of expansion. Human IntestiCult ODM human basal medium (ODM; 100-0212, StemCell Technologies; 500 μl, 50% v/v) and organoid supplement (100-019, StemCell Technologies) were added instead and supplemented with 100 μM l⁻¹ N-[N-(3, 5-difluorophenacetyl)-l-alanyl]-S-phenylglycinet-butyl ester (DAPT) (72082, StemCell Technologies). Differentiation media were changed every 2 d until usage for co-culture assay or fixation of the organoids.

### Intestinal organoid–PBMC co-culture
Intestinal organoids and tumouroids were cultivated and differentiated as described earlier. Three days after the initiation of differentiation, organoids and tumouroids were collected. Domes were washed in the plate with 500 μl cold 1X Dulbecco's phosphate-buffered saline (1X DPBS) (2345154, Gibco), followed by incubation with 500 μl cell recovery solution (35423, Corning) at 4 °C for 40 min. Next, organoid and tumouroid domes were disrupted by gentle up- and down-pipetting close to the dome. We then collected and transferred the suspension to a 15 ml Falcon tube, centrifuged it for 5 min at 200 g, followed by a wash with 'organoid wash' (see above for composition) and repeated centrifugation. Organoids and tumouroids were kept on ice until resuspension with PBMCs.

PBMCs were thawed following a standard thawing protocol. Briefly, PBMCs were thawed for 2 min in a 37 °C water bath. The cells were then transferred to a 15 ml Falcon tube. The PBMC vials were carefully rinsed with pre-heated 37 °C PBMC media (RPMI 1640 + Glutamax, 10% fetal bovine serum (heat inactivated) (97068-085, VWR), 1% penicillin/streptomycin (100X, 11140-035, Gibco), 1% non-essential amino acids (11360-039, Gibco, 100X), 1% sodium pyruvate (100X, 15140-122, Gibco), plus 1:100 DNAse (DNAse I, 0453628001, Roche)).

The suspension from the rinsed vial was added drop by drop to the cells in the 15 ml Falcon tube. Next, a further 6 ml of PBMC medium including 1:100 DNAse was added to the cells before centrifugation at 320 $g$ for 5 min. The supernatant was discarded before the cell pellet was resuspended in PBMC medium to yield ~4 million cells per ml. The resuspended cells were transferred to the incubator (37 °C, 5% CO$_2$) to rest for 15 min. Next, PBMCs were counted with Trypan Blue (216040, Invitrogen) on a Countess II (Invitrogen). After centrifugation at 320 $g$ for 5 min, PBMCs were then resuspended in PBMC medium at 4 million cells per ml and transferred in a 50 ml tissue culture flask (353014, Corning). The flask was placed slightly tilted in the incubator for a minimum of 4 h or overnight. After the incubation, PBMCs were transferred to a Falcon tube, centrifuged and stored on ice until resuspension with the intestinal organoids. Depending on the application, different matrices were generated.

In general, the number of PBMCs needed was calculated using the following formula: PBMCs needed = ($V$ (μl)×22,000)*$n$, where $V$ is the volume of dome and $n$ is the sample size. Immune-cell number was guided by the findings of Fu et al.[52] Dome volumes varied between imaging methods:

1. Killing assay: organoid–PBMC co-culture in a 100% Matrigel matrix for live imaging and fixed whole-mount imaging
   Domes of 10 μl per sample ($V$) were prepared to be seeded in a μ-Plate Angiogenesis 96-well black plate (69646, ibidi). First, PBMCs were resuspended in the necessary volume of 100% Matrigel. Second, the PBMC-matrix suspension was carefully mixed with the organoid pellet to decrease the likelihood of organoid damage. To avoid phase separation and Matrigel polymerization during the seeding process, the suspension was kept in a trough on ice. If multiple organoid lines were seeded simultaneously, organoid–PBMC suspensions were kept in PCR strips on ice to ease the seeding process, keeping the suspension cold, thus allowing continuous resuspension and faster handling using a multichannel pipette. After seeding the co-culture blend, the plates were flipped carefully. The flipped plate was placed in the incubator for 15 min. During the polymerization of the domes, PBMC media and ODM were prepared (50% v/v). Y27632 (10 mMol l$^{-1}$) was spiked into the blend. Media were added to the wells.
2. Organoid–PBMC co-culture in a 50% (v/v) collagen–Matrigel matrix for histology-based mIF imaging

For the co-culture setup ultimately embedded in FFPE blocks, 25 μl domes ($V$) of Matrigel and collagen (50% v/v) were prepared and seeded in a 24-well clear TC-treated plate. For the creation of the collagen–Matrigel matrix, 3D Culture Matrix rat collagen I (3447-020-01, R&D systems) was used. The collagen I was prepared by first using 1 M HEPES (15630-056, Gibco), followed by 37 mg ml$^{-1}$ NaMCO$_3$ (150 mM NaCl, 1 M HEPES) and finally collagen I at a ratio of 1:1:8. Matrigel and the collagen I mixture (50% v/v) were then mixed thoroughly. PBMC and organoid or tumouroid resuspension, dome polymerization and media preparation were executed as described in the previous paragraph.

Approximately 40 organoids per well was determined to yield the best results for the killing assay, both in terms of organoid segmentation during image analysis as well as statistical power, whereas ~100 organoids were seeded when performing the FFPE-based imaging approach in 25 μl domes. Exact organoid to PBMC cell number was confirmed by mIF quantification across dozens of samples at baseline (0 h condition). Analysis of the images yielded a 1.1:1 E:T ratio (considering EpCAM$^-$ cells as the entire PBMC population). Only considering the main effectors (CD4$^+$ and CD8$^+$ T cells) resulted in a 1:2.5 E:T ratio. Given the 10 μl dome for the killing assay, for example, this results in 220,000 PBMCs to ~200,000 epithelial cells, or 550,000 PBMCs to 500,000 epithelial cells in a 25 μl dome.

## T-cell bispecific antibodies for treatment
We previously provided detailed information about the construction of the CEA TCBs and NT TCB[53]. The trivalent 2 + 1 IgG EpCAM TCB was generated with the 'Knob-into-holes' technique, resulting in a monovalent CD3$^+$ lymphocyte arm as well as a bivalent target (EpCAM) engaging arm. All TCBs plus the corresponding NT TCB were titrated in PBMC–organoid media and applied 24 h post co-culture seeding, which denotes treatment day 0. Media were not changed after initiation of treatment.

## Blocking experiments
Where indicated, recombinant human monoclonal antibody etrolizumab (MA5-41929, Invitrogen; 10 μg ml$^{-1}$) and anti-integrin β7 monoclonal antibody (I-1141, Leinco; 10 μg ml$^{-1}$) were applied together with TCBs and re-titrated daily without changing the initial TCB-containing media.

## Low-resolution killing assay
Apoptosis was assessed using the CellEvent Caspase-3/7 detection reagent (C10423, Invitrogen) during the TCB-treatment period. CellEvent Caspase-3/7 detection reagent was diluted 1:1,000 and added to the respective master mix. Samples were imaged in confocal mode at ×5 magnification (air objective) with the Operetta CLS high-content imaging device (Perkin Elmer) covering a ~450 μm $Z$-stack, starting at −150 μm. The distance between $Z$-stacks was set to the minimum of 27 μm for the ×5 objective (autofocus: two peak; binning: 2). Channels selected were brightfield and the predefined Alexa 488. Per well, 5 fields were acquired, covering nearly the entire surface of the 96-well PhenoPlate. Given these imaging parameters, an imaging run of an entire 96-well plate took 2 h and 39 min. To rule out time-related killing impacts of TCBs applied between different plates, the time difference was accounted for by delaying TCB application per plate. The 72-h time course was imaged live every 24 h, starting with the 0 h baseline. CO$_2$ was set to 5%, temperature to 37 °C. Caspase-3/7 fluorescence signal intensity was quantified using the Opera Harmony software (Perkin Elmer). Briefly, segmentation of organoids was done by using 'Find Texture Regions' on the basis of the brightfield signal only, followed by 'Select Region' and 'Find Image Region' to segment single organoids as objects. Next, 'Calculate Morphology Parameters' was performed to select objects >7,500 μm$^2$ with 'Select Population'. Next, the caspase-3/7 fluorescence signal per individual organoid was determined using 'Calculate Intensity Properties' of the AF 488 channel within these objects.

## Flow cytometry
Five hours before collection, cultures were treated with eBioscience Protein Transport Inhibitor Cocktail (500X, ThermoFisher) to facilitate intracellular accumulation of temporally expressed soluble proteins. Duplicate wells of PBMC–organoid co-cultures from each condition were collected at 5, 24, 48 and 72 h post treatment. Co-cultures were first washed with PBS and then digested to single cells using Accutase solution. Cell suspensions were passed through a 70 μm strainer and stained for surface proteins. Cells were then fixed and permeabilized using the eBioscience Foxp3 Transcription Factor Staining Buffer set (ThermoFisher) and subsequently stained for intracellular and intranuclear proteins (Supplementary Table 2). Stained cell suspensions were acquired on a BD Fortessa X-20 flow cytometer (BD Biosciences) and analysed using FlowJo v.10.

## Analysis of cytokines
At each timepoint (0, 24, 48, 72 h) after TCB administration, 50 μl of supernatant was collected from the 24-well clear TC-treated plate. Supernatants were quickly centrifuged to remove cell debris and then frozen at −20 °C until further processing. Measurement of the cytokines (GM-CSF, IFNγ, TNFα, IL-2, IL-4, IL-6, IL-8, IL-10) was performed using the BioPlex Pro Human Cytokine 8-plex assay kit (M50000007A, Bio-Rad).

The controls, standards and beads were prepared by following manufacturer instructions. Briefly, we diluted the sample 1:2 according to guidelines. After preparing the 4-fold dilution series, we vortexed the beads and added them to the provided assay plate. After washing the plates with a plate washer (405TS, Bioteck), standards were added and incubated for 30 min at r.t. on a rocker at 850 r.p.m. (±50) as recommended. After another wash, detection antibodies were added, followed by incubation for 30 min at 850 r.p.m. before washing again. Lastly, we added the streptavidin-phycoerythrin mix, incubated the samples for 10 min at 850 r.p.m., ending with washing and resuspension of the beads with the provided assay buffer. The Luminex assay was performed with a BioPlex 200 instrument (Bio-Rad). First, the standard and control plates were measured, yielding a standard curve for each cytokine. Second, a quick validation of the concentration ranges confirmed the kit to be useful. Third, plates of interest were measured. With the previously generated best curve fit of the standards, the mean fluorescence intensity (MFI) of the samples was automatically calculated to the concentrations in pg ml$^{-1}$.

### FFPE-embedding of co-cultures
To FFPE-embed the co-cultures, the samples were seeded in a 50% (v/v) Matrigel–collagen I matrix as explained above. Wells were washed once with 1X DPBS before fixation with 4% paraformaldehyde (PFA) in the 24-well clear TC-treated plate. After 30 min of fixation at r.t., the wells were washed three more times before complete aspiration of the 1X DPBS. In the meantime, HistoGel (22-110-678, Thermo Scientific) was heated up according to manufacturer instructions. Pre-liquefied HistoGel (400 µl) was dispensed into the 24-well clear TC-treated plates. The plates were then placed in a 4 °C refrigerator for ~10 min. After polymerization of the HistoGel, the organoid–HistoGel 'platelet' was carefully lifted out of the 24-well clear TC-treated plate using a thin metallic spatula. If parts of the sample adhered to the plate, the collagen–Matrigel matrix was not of satisfactory quality. The samples were then distributed into biopsy cassettes and dehydrated overnight using a vacuum filter processor (Sakura, TissueTek VIP5). On the next day, the samples were embedded in liquid paraffin in metallic moulds and capped with the corresponding FFPE-block cassette. After polymerization of the paraffin, metallic moulds were removed and FFPE blocks stored at −20 °C until sectioning.

### Microtome sectioning
As the polymer structure of the embedded HistoGel differs substantially from that of the paraffin, organoid detection within the HistoGel was easily possible during sectioning. FFPE blocks were sectioned at a thickness of 3.5 µm and transferred on Superfrost Plus Adhesion microscope slides (J1800AMNZ, Epredia). Slides were incubated in a slide oven overnight at 37 °C, with the specimen facing up to prevent potential loss of organoids due to melting paraffin.

### H&E staining
H&E staining was executed in a fully automated manner following the standard protocol on the Ventana HE600 stainer (Roche Tissue Diagnostics).

### IHC staining
IHC staining of FFPE slides was performed using Ventana Discovery Ultra automated tissue stainer (Roche Tissue Diagnostics). Slides were baked first at 60 °C for 8 min and subsequently further heated up to 69 °C for 8 min for subsequent deparaffinization. This cycle was repeated three times. Heat-induced antigen retrieval was performed with Tris-EDTA buffer (pH 7.8; CC1, 950-227, Ventana) at 95 °C for 32 min. After blocking with Discovery Inhibitor (760-4840, Ventana) for 8 min, normal goat serum at 2% or 10% was applied as pre-treatment. Afterwards, primary antibodies (Supplementary Table 3) were diluted in Discovery Ab diluent (760-108, Ventana) and applied in the concentrations

determined in previous establishment runs. Primary antibodies were detected using anti-species secondary antibodies conjugated to horseradish peroxidase (HRP) (OmniMap anti-rabbit HRP, 760-4311; OmniMap anti-mouse, 760-4310, Ventana; OmniMap anti-rat, 760-4457, Ventana) for 16 min and subsequently visualized by conversion of 3,3'-diaminobenzidine (DAB) (Discovery ChromoMap DAB kit, 760-159, Ventana) or Discovery. Lastly, specimens were counterstained with haematoxylin (Haematoxylin II, 790-2208, Ventana) and bluing reagent (760-2037, Ventana). After dehydration with a standard series of alcohol (75%, 95%, 100%, 100% v/v; CAS64-17-5, Roche) and Xylol baths (100% v/v, 444240050, ACROS Organics), slides were mounted in a fully automated manner using the RCM7000 cover slipper (MEDITE) and a standard histoglue (00811-EX, Pertex). Slides were dried for at least 2 h before imaging.

### Brightfield imaging
HE and IHC stained slides were imaged with a brightfield whole-slide scanner at ×40 (Hamamatsu, NanoZoomer S360). Pixel size was 0.23 µm px$^{-1}$ in all brightfield images.

### Brightfield image analysis
Briefly, single organoids were automatically detected using a deep-learning algorithm trained to distinguish matrix and organoids or tumouroids (iterations: 1,030; cross-entropy: 0.08; DenseNet AI V2 Plugin). After quick validation, organoids and tumouroids were detected as objects and labelled as individual regions of interest (ROIs) (Supplementary Fig. 3b–d). Only objects >7,500 µm$^2$ were deemed positive. Area Quantification v.2.3.1 and Area Quantification FL v.2.2.2 modules were used to quantify positive-staining regions against overall size of each individual organoid and tumouroid, respectively (Supplementary Fig. 3b,d). Isotype controls and secondary-only negative controls on the tissue of origin served as a negative signal threshold to prevent biased adjustments.

### FFPE-based mIF staining
mIF staining of FFPE slides was performed using Ventana Discovery Ultra automated tissue stainer (Roche Tissue Diagnostics). Slides were first baked at 60 °C for 8 min and subsequently further heated up to 69 °C for 8 min for subsequent deparaffinization. This cycle was repeated twice. Heat-induced antigen retrieval was performed with Tris-EDTA buffer (pH 7.8; CC1, 950-227, Ventana) at 92 °C for 32 min. After each blocking step with Discovery Inhibitor (760-4840, Ventana) for 16 min, the Discovery Inhibitor was neutralized. Primary antibodies were diluted in 1X Plus Automation Amplification diluent (FP1609, Akoya Biosciences). Primaries were detected using anti-species secondary antibodies conjugated to HRP (OmniMap anti-rabbit HRP, 760-4311; OmniMap anti-mouse, 760-4310, Ventana; OmniMap anti-rat, 760-4457, Ventana) (Supplementary Table 3). Subsequently, the corresponding Opal dye (Opal 480, OP-001000; 520, OP-001001; 570, OP-001003; 620, OP-001004; 690, OP-001006; 780, OP-001008; Akoya Biosciences) was applied. After every application of a primary, corresponding secondary antibody and opal dye, an antibody neutralization and denaturation step was applied to remove residual antibodies and HRP, before starting the staining cycle again with the Discovery Inhibitor blocking step. Lastly, samples were counterstained with 4',6-diamidino-2-phenylindol (DAPI, Roche). Sequential order of the primary antibodies as well as corresponding dyes were determined during establishment runs. Neutralization of HRP and denaturation of the proteins were performed after every primary antibody cycle to avoid cross-bleeding and cross-reacting antibodies.

### FFPE-based mIF imaging
mIF stainings using the Opal dyes from Akoya were digitized using multispectral imaging by the Vectra Polaris (Perkin Elmer) employing the MOTiF technology at ×20 magnification for all 7 colours (Opal 480,

Opal 520, Opal 570, Opal 620, Opal 690, Opal 780 and DAPI). Laser exposure and intensity settings were adjusted on multiple slides per staining panel. Slides were scanned in a batch manner to ensure the same imaging settings and cross-comparability for later image analysis with HALO AI. Next, unmixing of the channels and tiling of the images were performed with PhenoChart (v.1.0.12) and inForm (v.2.4). Raw image data were saved as .qptiff. Tiles were fused in HALO (Indica Labs, v.3.2.1851.328). Pixel size in these images was 0.50 µm px$^{-1}$.

### FFPE-based mIF image analysis

Image analysis of the IF images was performed with HALO AI (Indica Labs, v.3.2.1851.328). Briefly, single organoids were automatically detected using a deep-learning algorithm trained to distinguish between matrix and organoids or tumouroids, (iterations: 11,415; cross-entropy: 0.428; DenseNet AI V2 Plugin). After quick validation, organoids and tumouroids were detected and labelled as individual ROIs, objects (Fig. 4c). Only objects >7,500 µm$^2$ were considered positive.

**Annotation layers and cell segmentation for immune cell quantification.** The ROI of the objects allowed generation of one inward and two outward concentric partitions (25 µm margins). To distinguish the four zones for subsequent analysis, the outer concentric outline was drawn as inclusion ROI (solid line), whereas the concentric outline defining the beginning of the next outline had to be copied and inverted as exclusive ROI (dotted line) into the corresponding annotation layer of the previous margin. This resulted in an annotation layer per zone, which allowed quantification of immune cells within (Fig. 3c and Supplementary Fig. 3f). The HighPlex FL v.4.1.3 module was used to perform nuclear segmentation on the basis of DAPI$^+$ cells. For detection and subsequent quantification, DAPI$^+$ cells and distinct markers for each individual cell type of interest were merged. Thus, secondary-only negative controls on the tissue of origin served as a negative signal threshold to prevent biased adjustments on the test slides. The analysis module was deployed on ROIs per single object (organoid or tumouroid). Where indicated, normalization to object area was performed. Overlapping organoid and tumouroid zones were deleted.

**Quantification of E-cadherin$^+$, panCK$^+$ and Caspase-3$^+$ epithelium.** Quantification of TCB-triggered epithelial killing via increase in caspase-3$^+$ (apoptosis) and decrease in panCK$^+$ epithelium (loss of integrity) of the organoids and tumouroids was performed by harnessing the DenseNet AI V2 classifier trained to distinguish the matrix, organoids, caspase-3$^+$ and the panCK$^+$ signal in the 25 µm margin of epithelium of both organoids and tumouroids (Supplementary Fig. 3e; iterations: 34,035; cross-entropy: 0.5; HALO AI). For each class, the area covered was quantified and calculated against the overall area of epithelium of the objects, yielding distinct percentages.

### Whole-mount immunofluorescence staining

Before fixation with 4% PFA (43368, Alfa Aesar) for 30 min at r.t., samples were washed with 1X DPBS (14287-050, Gibco). After fixation, samples were washed three more times with 1X DPBS on a rocker for 15 min at 150 r.p.m. Next, 1X DPBS was removed and samples were stained with DAPI (1:2,000 dilution; 62248, Thermo Scientific), Phalloidin-Atto 594 (1:3,000 dilution; 51927, Sigma) and Caspase-3/7 (1:1,000 dilution; C10423, Invitrogen). After 90 min incubation at r.t., samples were washed three times with 1X DPBS, again using a rocker. Images were acquired with the Operetta CLS (Perkin Elmer) using a ×20 water objective. Optical mode, focus and binning were set up as described above. The first plane imaged was at −15 µm, followed by 50 z-stacks with a distance of 4 µm. 3D reconstruction was performed in Harmony (Perkin Elmer).

### Statistics

Details about statistical analysis are provided in each figure's description. Unpaired t-tests were applied to compare two datasets. One-way and two-way analysis of variance (ANOVA) including Tukey's multiple comparison tests were used for more than two datasets. Datasets are visualized with mean and standard deviation as indicated in the figure legends, unless documented otherwise. P values to indicate statistical significance are indicated in the figures. ROUT outlier analysis with low Q coefficients in GraphPad Prism 8 (GraphPad Software) was performed to remove definite outliers, as indicated in figure captions. All graphs in the document were generated in GraphPad Prism 8.

### Reporting summary

Further information on research design is available in the Nature Portfolio Reporting Summary linked to this article.

### Data availability

The main data supporting the results in this study are available within the paper and its Supplementary Information. Source data for the figures are provided with this paper. The raw and analysed datasets generated during the study are available for research purposes from the corresponding author on reasonable request.

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

## Acknowledgements

We thank R. Sriram for supporting us with the establishment and oversight of the research agreements governing access to human intestinal specimens; A. Bransi and his team for the production of EpCAM TCB. We acknowledge the support of the non-profit foundation HTCR, which holds human tissue on trust, making it broadly available for research on an ethical and legal basis.

## Author contributions

M.F.H., N.G., N.S.-R. and A.S. conceived the study. M.F.H. and N.G. wrote the manuscript. M.F.H. was involved in designing and performing all experiments in the manuscript. T.R. and K.K. performed flow-cytometry experiments. R.G. and B.A. established an early version of the organoid–PBMC co-culture assay and strategies for analysis. Y.B. and J.B. helped design and perform revision experiments. C.E., K.G.-J, S.P. and A.F. generated and expanded organoids and tumouroids. J.A. and M.F.H. established protocols and devices for high-throughput sectioning of organoid–PBMC co-cultures. M. Bscheider established the research agreements with HTCR providing access to human intestinal specimens. M. Bacac and her team provided reagents (CEA-targeting TCBs). All authors provided feedback on the manuscript.

## Competing interests

All authors, except for C.E., K.G.-J and S.P., are employees of Hoffmann-LaRoche. The company provided support in the form of salaries for authors, but did not have any additional role in the study design, data collection and analysis, decision to publish, or preparation of the manuscript. C.E., K.G.-J and S.P. declare no competing interests.

## Additional information

**Extended data** is available for this paper at https://doi.org/10.1038/s41551-023-01156-5.

**Correspondence and requests for materials** should be addressed to Nikolche Gjorevski.

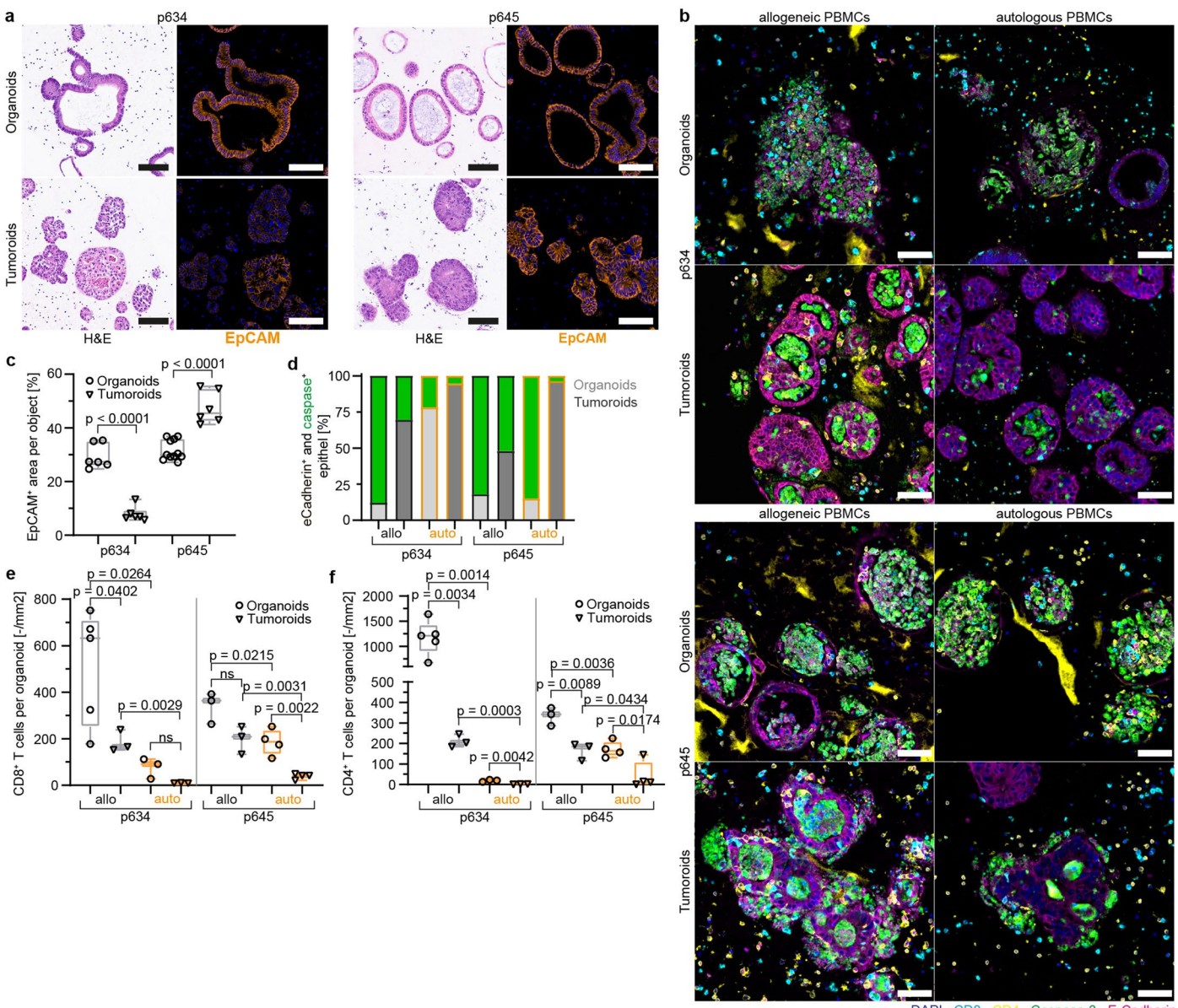

**Extended Data Fig. 1 | TCB-mediated apoptosis and T-cell infiltration across different donor-matched organoid–tumour pairs. a**, Comparison of morphology and target antigen expression between organoids and tumouroids by H&E and IHC at 20x. Scale: 100 µm. **b**, Representative images at 20x magnification of both donor-matched pairs 48 h post EpCAM TCB treatment (10 µg/mL), displaying E-cadherin+ organoids and tumouroids (magenta) surrounded by CD4+ (yellow) and CD8+ (turquoise) T cells. Caspase-3 (green) captures TCB-triggered immune-induced apoptosis, nuclei were counterstained with DAPI (blue). Scale: 100 µm. **c**, Quantification of EpCAM expression by positive area per individual organoid, plotted as boxplot, whiskers showing all

points (min to max; n ≥ 6). Statistical analysis was performed by unpaired t-tests (two-tailed) and was defined as ****p < 0.0001. **d**, Sum of E-cadherin+ (grey) and caspase-3+ (green) epithelium of the organoids and tumouroids, respectively, detected in zone 2 (on epithelium) 48 h post EpCAM TCB treatment (n = 3). **e** and **f**, Allogeneic (allo) and autologous (auto) CD8+ T cell infiltration (panel e) and CD4+ T cell infiltration (panel f) per organoid and tumouroid (normalized for area of objects) as boxplot, whiskers showing all points (min to max; n ≥ 3). Statistical analysis was conducted by an unpaired t-test (two-tailed) and was defined as *p < 0.05, **p < 0.01. The displayed experiment in this figure was performed once.

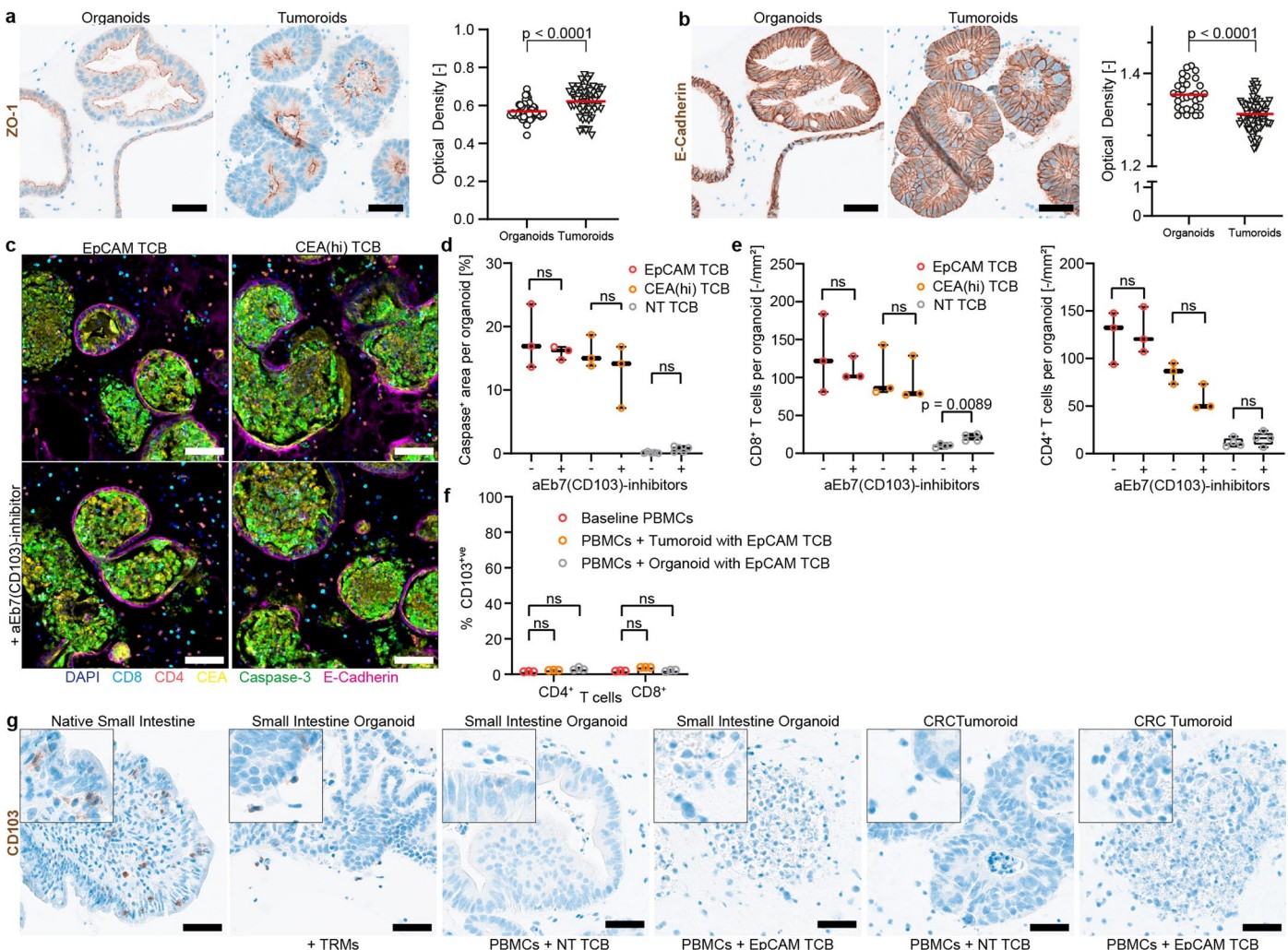

**Extended Data Fig. 2 | TCB-triggered T-cell infiltration is not mediated by CD103-E-cadherin interactions. a** and **b**, Chromogenic staining (brown) with according quantification of tight junctions (ZO-1; $n_{OrgZO-1}$ = 59, $n_{TumZO-1}$ = 58) and adherent junctions (e-Cadherin; $n_{OrgZO-1}$ = 31, $n_{TumZO-1}$ = 84) in organoids and tumouroids. 20x magnification, scale: 100 μm. Statistical analysis was conducted by an unpaired t-test (two-tailed) and was defined as ****p < 0.0001. Red line displays mean. **c**, Representative images at 20x magnification of organoids treated with EpCAM and CEA(hi) TCB (1 μg/mL) at 48 h. Top row displays TCBs only, bottom row is co-treated with an αE(CD103)β7-inhibitor cocktail (Etrolizumab (10 μg/mL) and an anti-Integrin β7 monoclonal antibody (10 μg/mL)). E-cadherin⁺ organoids (magenta) surrounded by CD4⁺ (orange), CD8⁺ (turquoise) and CEA⁺ (yellow) immune cells. Caspase-3 (green) captures TCB-triggered immune-induced apoptosis, nuclei are stained with DAPI (blue). Scale: 100 μm. **d**, Quantification of caspase⁺ area per organoid across different TCB treatments (48 h) with (+) and without (-) an αE(CD103)β7-inhibitor cocktail, plotted as boxplot, whiskers showing all points (min to max; $n_{NT TCB}$ = 5;

$n_{EpCAM TCB}$ = 3; $n_{CEA TCB}$ = 3). Statistical analysis was conducted by an unpaired t-test (two-tailed) and was found to be non-significant (ns). **e**, Quantification of T cell infiltration per organoid area across different TCB treatments (48 h) with (+) and without (-) an αE(CD103)β7-inhibitor cocktail, plotted as boxplot, whiskers showing all points (min to max; $n_{NT TCB}$ = 4; $n_{EpCAM TCB}$ = 3; $n_{CEA TCB}$ = 3). Statistical analysis was performed by unpaired t-tests (two-tailed) and was defined as **p < 0.01 and non-significant (ns) (ns). **f**, FACS staining of CD103 on organoid and tumouroid co-cultured with PBMCs, treated with EpCAM TCB for 72 h. Data are presented as mean values ∓ SD (n = 3). Statistical analysis was conducted by an unpaired t-test (two-tailed) and was found to be non-significant (ns). **g**, Chromogenic DAB staining of CD103 (brown) in healthy small intestine tissue and different organoid-immune co-cultures, scanned at 20x. Scale: 100 μm. CD103⁺ immune cells only found in native SI tissue and expressed by tissue-resident memory T cells (TRMs). The displayed experiment in this figure was performed once.

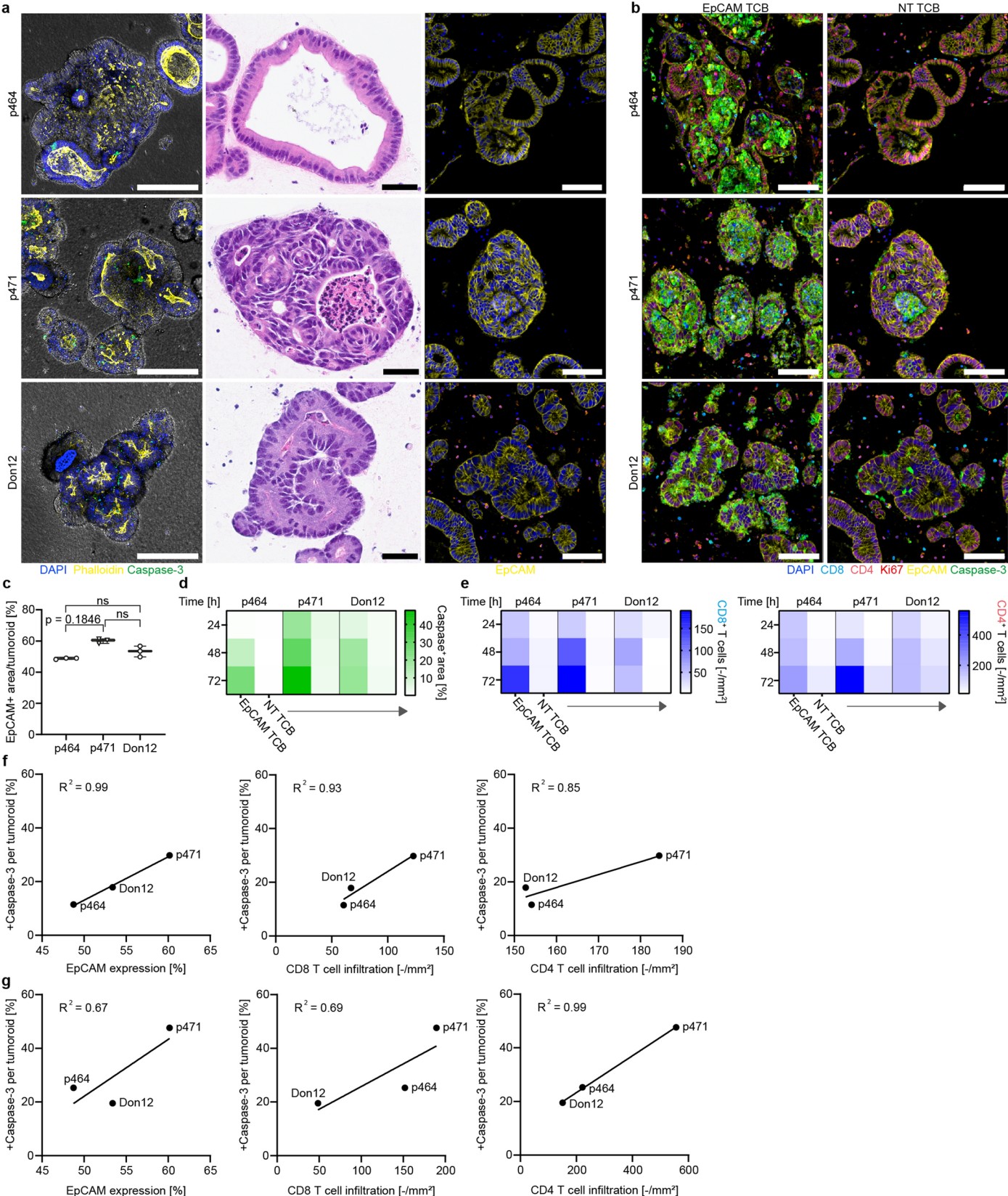

**Extended Data Fig. 3 | See next page for caption.**

**Extended Data Fig. 3 | Target expression, rather than tumouroid morphology, governs TCB-induced damage and T-cell infiltration. a**, F-actin⁺ (Phalloidin) outlined tumouroids (yellow) stained with Caspase-3/7 (green) and DAPI⁺ (blue displayed as maximum intensity projection of a z-stack of around 100 μm. 20x magnification, scale: 200 μm. H&E highlights architectural differences between tumouroid donors at 40x magnification. Scale: 50 μm. Expression of target antigen EpCAM⁺ for each tumouroid donor. Imaged at 20x, scale: 100 μm. **b**, Representative images tumouroids 72 h post EpCAM and NT TCB treatment (10 μg/mL). Staining for target EpCAM⁺ (yellow) highlights highly Ki67⁺ (red) tumouroids surrounded by CD4⁺ (orange) and CD8⁺ (turquoise) T cells. Caspase-3 (green) captures TCB-triggered immune-induced apoptosis, nuclei were counterstained with DAPI (blue). Scale: 100 μm. **c**, Quantification of EpCAM expression by positive area per individual organoid, plotted as mean

per replicate (n = 3). Statistical analysis was performed by unpaired t-tests (two-tailed) and was defined as *p < 0.05 and non-significant (ns). **d**, Representative images at 20x magnification of both donor-matched pairs 48 h post EpCAM TCB treatment (10 μg/mL), displaying E-cadherin⁺ organoids and tumouroids (magenta) surrounded by CD4⁺ (yellow) and CD8⁺ (turquoise) T cells. Caspase-3 (green) captures TCB-triggered immune-induced apoptosis, nuclei were counterstained with DAPI (blue). Scale: 100 μm. **d**, Quantification of caspase⁺ area per organoid treated with EpCAM and NT TCB (n = 3). **e**, CD8⁺ T cell infiltration and CD4⁺ T cell infiltration per tumouroid, plotted as mean (n = 3; normalized for area of objects). **f** and **g**, Correlation plot between caspase-3⁺ expression, CD4⁺ and CD8⁺ versus EpCAM expression per tumouroid at 48 h post (**f**) and 72 h post EpCAM TCB treatment (**g**). The displayed experiment in this figure was replicated once, yielding similar results.

# Reporting Summary

## Statistics

For all statistical analyses, confirm that the following items are present in the figure legend, table legend, main text, or Methods section.

| n/a | Confirmed | |
|---|---|---|
| ☐ | ☒ | The exact sample size (*n*) for each experimental group/condition, given as a discrete number and unit of measurement |
| ☐ | ☒ | A statement on whether measurements were taken from distinct samples or whether the same sample was measured repeatedly |
| ☐ | ☒ | The statistical test(s) used AND whether they are one- or two-sided *Only common tests should be described solely by name; describe more complex techniques in the Methods section.* |
| ☒ | ☐ | A description of all covariates tested |
| ☒ | ☐ | A description of any assumptions or corrections, such as tests of normality and adjustment for multiple comparisons |
| ☐ | ☒ | A full description of the statistical parameters including central tendency (e.g. means) or other basic estimates (e.g. regression coefficient) AND variation (e.g. standard deviation) or associated estimates of uncertainty (e.g. confidence intervals) |
| ☐ | ☒ | For null hypothesis testing, the test statistic (e.g. *F*, *t*, *r*) with confidence intervals, effect sizes, degrees of freedom and *P* value noted *Give P values as exact values whenever suitable.* |
| ☒ | ☐ | For Bayesian analysis, information on the choice of priors and Markov chain Monte Carlo settings |
| ☒ | ☐ | For hierarchical and complex designs, identification of the appropriate level for tests and full reporting of outcomes |
| ☒ | ☐ | Estimates of effect sizes (e.g. Cohen's *d*, Pearson's *r*), indicating how they were calculated |

*Our web collection on statistics for biologists contains articles on many of the points above.*

## Software and code

Policy information about availability of computer code

| Data collection | No software was used for data collection. |
|---|---|
| Data analysis | Commercial software packages used for data analysis: Mage analysis: HALO AI v3.4.2986.209 (Indica Labs) and Harmony v4.9 (Perkin Elmer) Statistical analysis: GraphPad Prism v8.4.2 Flow-cytometry data analysis: FlowJo v10.8.1 |

For manuscripts utilizing custom algorithms or software that are central to the research but not yet described in published literature, software must be made available to editors and reviewers. We strongly encourage code deposition in a community repository (e.g. GitHub). See the Nature Portfolio guidelines for submitting code & software for further information.

## Data

Policy information about [availability of data](availability of data)

All manuscripts must include a [data availability statement](data availability statement). This statement should provide the following information, where applicable:
- Accession codes, unique identifiers, or web links for publicly available datasets
- A description of any restrictions on data availability
- For clinical datasets or third party data, please ensure that the statement adheres to our [policy](policy)

> The main data supporting the results in this study are available within the paper and its Supplementary Information. Source data for the figures are provided with this paper. The raw and analysed datasets generated during the study are available for research purposes from the corresponding authors on reasonable request.

## Human research participants

Policy information about [studies involving human research participants and Sex and Gender in Research.](studies involving human research participants and Sex and Gender in Research.)

| | |
|---|---|
| Reporting on sex and gender | The experiments reported in the paper were performed using intestinal resections derived from a nearly equal number of male (11) and female (8) donors (wherein 'male' and 'female' refers to sex) (Supplementary Table 1). Information on the gender of the donors was not available and we are unable to report it. We did not observe sex to have any discernible influence on the main readouts within the study (target expression, TCB-mediated apoptosis/cytokine release/T-cell infiltration), which were strictly governed by variables independent of sex (such as intestinal region, TCB type and concentration). |
| Population characteristics | The information on donor age is included in Supplementary Table 1. Small intestinal donors had been diagnosed with pancreatic cancer or pancreatitis, whereas colon and rectum donors had been diagnosed with colorectal cancer. Genotypic information on driver mutations and treatment history is available on request, but not included in this paper as we deem it not relevant to the main aims and conclusions. |
| Recruitment | We did not devise a prospective recruitment strategy. We included into the study any available surgical specimen of human intestinal tissue during the period over which the study was performed. |
| Ethics oversight | The framework of the HTCR Foundation, which includes written informed consent from all donors, was approved by the ethics commission of the Faculty of Medicine in the LMU (number 025-12) and the Bavarian State Medical Association (number 11142). |

Note that full information on the approval of the study protocol must also be provided in the manuscript.

# Field-specific reporting

Please select the one below that is the best fit for your research. If you are not sure, read the appropriate sections before making your selection.

☒ Life sciences  ☐ Behavioural & social sciences  ☐ Ecological, evolutionary & environmental sciences

For a reference copy of the document with all sections, see [nature.com/documents/nr-reporting-summary-flat.pdf](nature.com/documents/nr-reporting-summary-flat.pdf)

# Life sciences study design

All studies must disclose on these points even when the disclosure is negative.

| | |
|---|---|
| Sample size | No statistical methods were used to predetermine sample sizes. Sample sizes used were deemed to be representative and sufficient, on the basis of the low variability of the results between technical and biological replicates and on overall reproducibility. |
| Data exclusions | No data were excluded. |
| Replication | All experiments presented in this study were repeated at least three times (most of them more than five times) to ensure the reproducibility of the results. All attempts at replication were successful. |
| Randomization | Randomization was not performed, because we did not identify factors that would co-vary with different treatment conditions within a single experiment. |
| Blinding | Blinding was not performed, as all analyses (flow cytometry, cytokine analysis, image analysis) were performed in bulk, applying the exact same settings for all treatment conditions. |

# Reporting for specific materials, systems and methods

We require information from authors about some types of materials, experimental systems and methods used in many studies. Here, indicate whether each material, system or method listed is relevant to your study. If you are not sure if a list item applies to your research, read the appropriate section before selecting a response.

## Materials & experimental systems

| n/a | Involved in the study |
|-----|-----------------------|
| ☐ | ☒ Antibodies |
| ☐ | ☒ Eukaryotic cell lines |
| ☒ | ☐ Palaeontology and archaeology |
| ☒ | ☐ Animals and other organisms |
| ☒ | ☐ Clinical data |
| ☒ | ☐ Dual use research of concern |

## Methods

| n/a | Involved in the study |
|-----|-----------------------|
| ☒ | ☐ ChIP-seq |
| ☐ | ☒ Flow cytometry |
| ☒ | ☐ MRI-based neuroimaging |

## Antibodies

| | |
|---|---|
| Antibodies used | 1. IHC Antibodies: Antigen, Vendor, Clone, Dilution, Reference;<br>Recombinant SOX9, Abcam, EPR14335-78, 800, Ab1859666; FABP1, Life Technologies, N/A, 100, PA5-28945; MUC2, Life Technologies, 996/1, 100, MA5-12345; Ki67, Invitrogen, SolA15, 500, 14-569882; CD3, Ventana, 2GV6, prediluted, 790-4341; CD4, Ventana, SP35, prediluted, 790-4423; CD8, Ventana, SP57, prediluted, 790-4460; CD14, Abcam, 1H5D8, 200, ab181470; CD20, DAKO, L26, 200, M0755; Recombinant CD103, Abcam, SP301, 100, Ab227697; Cleaved Caspase-3, Cell Signaling Technology, Asp175, 100, 9661; CEA, Abcam, EPR20721, 100, ab207718; EpCAM, Ventana, Ber-EP4, prediluted, 760-4383; E-Cadherin, Ventana, 36, prediluted, 760-4497; Pan-Cytokeratin, Biorybt, C-11, 100, Orb43707; Granzyme B, Abcam, 100, Ab4059; ZO-1, Invitrogen, Z11A12, 50, 33-9100<br><br>2. Flow-cytometry antibodies: Antigen, Vendor, Clone, Fluorophore, Dilution, Reference;<br>IIFNγ, BD Biosciences, B27, BUV395, 200, 563563; CD69, BD Biosciences, FN50, BUV737, 200, 612817; CD8, BD Biosciences, SK1, BUV805, 200, 612889; Ki-67, Biolegend, Ki-67, BV421, 200, 350506; CD103, Biolegend, Ber-ACT8, BV421, 200, 350214; CD45, Biolegend, 2D1, BV510, 200, 368526; CD19, Biolegend, HIB19, BV605, 200, 302244; HLA-DR, Biolegend, L243, BV650, 200, 307650; CD11b, Biolegend, ICRF44, BV711, 200, 301344; CD45RA, Biolegend, HI100, BV786, 200, 304140; CD45RO, Biolegend, UCHL1, FITC, 200, 304204; TNFα, Biolegend, MAb11, PE, 200, 502909; Granzyme B, Biolegend, QA16A02, PE-Dazzle-594, 200, 372215; 41BB, Biolegend, 4B4-1, PE-Cy7, 200, 309818; IL-2, Biolegend, MQ1-17H12, APC, 200, 500310; CD3, Biolegend, HIT3a, Alexa Fluor 700, 200, 300324; Efluor780 Fixable L/D, Thermo, N/A, APC-H7, N/A, 65-0865-14 |
| Validation | Secondary only and isotypes controls were applied during establishments of antibody staining. All primary antibodies were tested on human native tissue as validation for specific staining, and evaluated by a pathologist (Nadine Stokar) before applying them on organoids. |

## Eukaryotic cell lines

Policy information about cell lines and Sex and Gender in Research

| | |
|---|---|
| Cell line source(s) | Organoid cell lines were derived from primary surgical resections. Healthy organoids were derived from the healthy margin of tumour resections, whereas tumour organoids were derived from the malignant tissue. Information on the sex of the patients is available for each organoid line (Supplementary Table 1). |
| Authentication | Histological assessment and IHC staining for intestinal markers were performed by a trained pathologist to verify the intestinal epithelial origin of the organoids, and to verify the classification into healthy vs. transformed epithelium. |
| Mycoplasma contamination | All cell lines used were verified to be negative for mycoplasma before experimentation. |
| Commonly misidentified lines (See ICLAC register) | No commonly misidentified cell lines were used. |

## Flow Cytometry

### Plots

Confirm that:

☒ The axis labels state the marker and fluorochrome used (e.g. CD4-FITC).

☒ The axis scales are clearly visible. Include numbers along axes only for bottom left plot of group (a 'group' is an analysis of identical markers).

☒ All plots are contour plots with outliers or pseudocolor plots.

☒ A numerical value for number of cells or percentage (with statistics) is provided.

### Methodology

| | |
|---|---|
| Sample preparation | PBMCs-organoid cocultures were digested into a single-cell suspension with accutase and filtered, subjected to flow cytometry staining (as detailed in Methods) and then acquired immediately |

| | |
|---|---|
| Instrument | BD LSRFortessa X20 |
| Software | FowJo v10.8.1 |
| Cell population abundance | No cell populations were sorted. |
| Gating strategy | Provided in Supplementary Fig. 2. |

☒ Tick this box to confirm that a figure exemplifying the gating strategy is provided in the Supplementary Information.

