## [Peer Review File · Nature Biomedical Engineering]

Analysis of off-tumour toxicities of T-cell-engaging bispecific antibodies via donor-matched intestinal organoids and tumouroids

Corresponding author: Nikolche Gjorevski

Editorial note

This document includes relevant written communications between the manuscript's corresponding author and the editor and reviewers of the manuscript during peer review. It includes decision letters relaying any editorial points and peer-review reports, and the authors' replies to these (under 'Rebuttal' headings). The editorial decisions are signed by the manuscript's handling editor, yet the editorial team and ultimately the journal's Chief Editor share responsibility for all decisions.

Any relevant documents attached to the decision letters are referred to as **Appendix #**, and can be found appended to this document. Any information deemed confidential has been redacted or removed. Earlier versions of the manuscript are not published, yet the originally submitted version may be available as a preprint. Because of editorial edits and changes during peer review, the published title of the paper and the title mentioned in below correspondence may differ.

This manuscript was reconsidered after the authors' response (available as 'Rebuttal 1' in this document) to the editor's initial decision on 07 Oct 2022.

Correspondence

Fri 07 Oct 2022

Decision on Article nBME-22-2306

Dear Dr Gjorevski,

Thank you for submitting to *Nature Biomedical Engineering* your manuscript, "Organoid model to predict and study immunotherapy-induced damage of healthy and tumor tissue". I regret that we are unable to consider it for external peer review.

As you may know, we screen manuscripts against our editorial criteria, and decline a substantial proportion of them without input from external referees (pertinent journal-level statistics are available). In such cases, even if reviewers were to certify the manuscript as technically correct, we feel that it would not be of outstanding interest to merit publication in *Nature Biomedical Engineering*. These editorial judgements are based on considerations of the degree of advance, broad implications, and breadth and depth of the work reported in the manuscript.

The topic of the manuscript is within the remit of the journal, and we do acknowledge your efforts in achieving immune competent cancer organoids that allowed the studying of the cytotoxicity of bispecific antibodies. However, in view of published work on immune competent cancer mimickry cultures (including microphysiological systems), and in the absence of sufficient evidence demonstrating that the side effects observed in humans are caused by cytotoxicity on patient-derived cells healthy epithelial cells, and because an extensive molecular and functional characterization of the organoids is lacking, we feel that this study does not reach the high threshold in technological, translational, mechanistic and pre-clinical advances that we look for in comparable manuscripts that we consider for external peer review.Nevertheless, the work may be appropriate for Communications Biology (a selective open-access journal from Nature Portfolio), to which it could be transferred by using the link in the footnote below.

We appreciate that rejections are unwelcome. To make them easier to understand, we wrote about how we decline manuscripts.

Best wishes,

Valeria

Dr Valeria Caprettini
Associate Editor, Nature Biomedical Engineering

* We recommend that you consider *Communications Biology* as a suitable journal for your work. To transfer your manuscript to the journal, please use our manuscript transfer portal. You will not have to re-supply manuscript metadata and files, unless you wish to make modifications. This link can only be used once and remains active until used. For more information, please see our manuscript transfer FAQ page.

Note that any decision to opt in to *In Review* at the original journal is not sent to the receiving journal on transfer. You can opt in to In Review at receiving journals that support this service by choosing to modify your manuscript on transfer. *In Review* is available for primary research manuscript types only.

Tue 14 Feb 2023

Decision on Article NBME-22-2306A-Z

Dear Dr Gjorevski,

Thank you again for submitting to *Nature Biomedical Engineering* your manuscript, "Organoid model to predict and study immunotherapy-induced damage of healthy and tumor tissue" and thank you for your patience in waiting our response. The manuscript has been seen by 3 experts, whose reports you will find at the end of this message.

You will see that the reviewers appreciate the work. However, they express concerns about the degree of support for the claims, and provide useful suggestions for improvement. We hope that with significant further work you can address the criticisms and convince the reviewers of the merits of the study. In particular, we would expect that a revised version of the manuscript provides:

- * Further discussion on the limitation of the model for predicting off-tumour toxicities as per reviewer #1 suggestion.
- * Discussion and clarification on how many organoid tumoroid pairs have been used, how many different donors are the PBMCs coming from, and other details as Reviewer #2 requested.
- * Quantification of the degrees of heterogeneity of the organoids and how these affect the findings, as directly or indirectly queried by Reviewers #1 and #2.
- * Additional assessment of the reasons for the different sensitivities of the organoids to antibody dose and affinity, as requested by Reviewer #2.
- * Thorough methodological reporting, as per the relevant requests of all reviewers.

When you are ready to resubmit your manuscript, please upload the revised files, a point-by-point rebuttal to the comments from all reviewers, the reporting summary, and a cover letter that explains the main improvements included in the revision and responds to any points highlighted in this decision.

Please follow the following recommendations:

- * Clearly highlight any amendments to the text and figures to help the reviewers and editors find and understand the changes (yet keep in mind that excessive marking can hinder readability).
- * If you and your co-authors disagree with a criticism, provide the arguments to the reviewer (optionally, indicate the relevant points in the cover letter).
- * If a criticism or suggestion is not addressed, please indicate so in the rebuttal to the reviewer comments and explain the reason(s).
- * Consider including responses to any criticisms raised by more than one reviewer at the beginning of the rebuttal, in a section addressed to all reviewers.
- * The rebuttal should include the reviewer comments in point-by-point format (please note that we provide all reviewers will the reports as they appear at the end of this message).
- * Provide the rebuttal to the reviewer comments and the cover letter as separate files.

We hope that you will be able to resubmit the manuscript within 20 weeks from the receipt of this message. If this is the case, you will be protected against potential scooping. Otherwise, we will be happy to consider a revised manuscript as long as the significance of the work is not compromised by work published elsewhere or accepted for publication at *Nature Biomedical Engineering*.

We hope that you will find the referee reports helpful when revising the work. Please do not hesitate to contact me should you have any questions.

Best wishes,

Valeria

Dr Valeria Caprettini
Associate Editor, Nature Biomedical Engineering

Reviewer #1 (Report for the authors (Required)):

In this manuscript by Harter et al describe an organoid based platform to predict on-target off-tumor intestinal toxicities of T-cell bispecific antibodies (TCBs) targeting EpCAM and CEA through co-culture of patient-derived intestinal organoids and allogenic PBMC. By using matched healthy and tumor organoids, they showed a potential to measure combined efficacy and toxicity of TCBs. With this model, they showed mechanism of epithelial damage by T-cells activation cascade and subsequent Granzyme B production through FACS analysis, and spatiotemporal features of organoids-immune cells interaction by using a 7-plex mIF method. The co-culture model is described by other group already, however, the application for prediction of on-target off-tumor intestinal toxicities of immunotherapy seems advances in the aspect of clinical/translational application of the co-culture model.

Major comments:

1. The authors described inter-donor variability of TCB toxicities, however, they did not describe inter-PBMC variability. To recapitulate clinical responses and toxicities, autologous PBMC and healthy / tumor organoids co-culture models from cancer patients seems more appropriate.
2. A limitation of the co-culture model is not applicable for predicting off-tumor toxicities of immune checkpoint inhibitors. Limitations of this models for predicting off-tumor toxicities of T-cell targeted approach should be mentioned.
3. Unlikely to healthy organoids, or tumoroids morphology is not uniform as described in other reports. In this manuscript, the authors did not mention the number of combined safety-efficacy experiments. Considering intertumoral heterogeneity, several additional experiments using tumoroids of other morphology potentiate the claim of combined safety-efficacy profiling platform.

Minor comments:

1. Inter-personal variability may influence growth of organoids and eventually affect to organoid differentiation, morphology and target expression. As shown in figure 1b and figure 4a, individual organoids in a single culture also differ in target expression. If TCB toxicities and efficacy are related to amount of target expression, mIF can visualize more T-cell infiltration in organoids with higher target expression than low target expression in a single experiment.
2. In 96-well plate seeding, 15,000 PBMCs and an unknown number of organoids are co-cultured in a 5ul Matrigel dome/well. If the number of cells is greater than a proper amount of Matrigel, there may be difficulty in Matrigel hardening and a risk of dome detachment from culture plate during media change and culturing. Precise description regarding the ratio of organoids/PBMC cell number and total cell number are required.
3. Due to the sticky nature of Matrigel, technical problem such as air bubble formation and phase separation easily occur during resuspension process. For this reason, mentioning of success rate and total number of co-culture and TCB treatment experiments are required.
4. Display of data in Fig 5 is complicating. Sequences of donor organoids number in Fig 5a, b, c, d is inconsistent.
5. Lane #246 Fig. 3a is not matched with the text.

Reviewer #2 (Report for the authors (Required)):

Harter et al describe the use of human intestinal epithelial organoids from normal small and large bowel mucosa as in vitro models to assess the toxicity of immunotherapies that target cell surface antigens. This model is novel and highly important as animal models that recapitulate the expression of relevant human antigens are often lacking. This is hence a major new addition to the repertoire of tests that can be used to define potential safety issues with new immunotherapeutics before testing these in patients. The following points need to be addressed before publication is considered:

Major criticism:

Line 286: "high ZO-1 expression might be considered an immune-protective, whereas E287 cadherin expression an immune-inclusive feature"

I think this is too simplistic. ZO-1 expression needs to be in the right place to hinder infiltration and E-cadherin is not predominantly an immune regulatory molecule but has a broad range of functions. Rephrasing would be useful.

Figure 4f, g: the staining pattern further confirms my concern above. ZO-1 is for example mainly expressed on the luminal side of the organoids where it is unlikely to impair T cell infiltration. And although the optical density of E cadherin is numerically lower in cancer cells, it still stains strongly in these, questioning whether this difference really has a relevant impact. I think you would need blocking experiments to assess whether E cadherin expression is important for infiltration. It is also not clear how many different organoid tumoroid pairs were used in these experiments. In order to make general statements about differences in organoid and tumoroid, this comparison needs to be done in a good number of these. The same applies for the statement "Given the striking differences in epithelial architecture and morphology between the two tissues types (Fig. 4a)". You need to show this across a reasonable number of organoid-tumoroid pairs.

Line 299: Quantification of TCB-mediated organoid killing revealed striking differences in response within the organoid cohort (Fig. 5a,b), wherein certain organoid lines were targeted readily and others seemed refractory to damage.

A key question here is how many different donors were used for the PBMC generation and whether the striking differences you mention were observed in experiments done at the same time and with the same donor. How many repeats were performed? The lack of this kind of detailed information makes the rigidity of the experimental approach difficult to judge. When discussing results from Fig 5, a more quantitative approach would be useful as the description is rather vague as it stands. What proportion of organoids showed what degree of damage?

In the final paragraph of the results section, I would have thought that it makes sense to assess reasons for different sensitivities in more detail, for example with by performing the same FACS analyses (as shown in Fig 2) to assess T cell activation status and effector functions and by 3D multiplex immunostaining (as shown in Fig 3) to investigate whether this is due to a lack of T cell infiltration.

Discussion: You show that small intestine and colon organoids differ in EPCAM and CEA expression but it is not mentioned whether this now correlates with small or large bowel toxicities that were observed for such therapies in the clinic. Did CEA targeting drugs mainly trigger large bowel toxicities and does this match observations in your models? I am not sure how much of this clinical trial data has been published but the issue deserved at least some discussion.

I understand that the surgical specimens were obtained from a commercial provider but the methods section should nevertheless provide information on how ethics approval for their use was obtained and whether patients provided written informed consent.

Minor criticism:

Line 52: The authors should also cite Gonzalez-Exposito JITC 2019 and Schnalzger Embo J 2019 as these were the first manuscripts showing the feasibility of 3D organoid + T cell co-culture to assess novel antigen specific therapeutics in bowel cancers.

A table showing the number of organoids, the anatomical origin, and basic donor characteristics (age, sex) should be included in the supplements.

111: The solid extracellular matrix (ECM) surrounding the organoids and PBMCs is a more faithful representation of the native tissue context and allows simulating crucial immunological processes beyond contact-dependent targeting, including bystander signaling, immune cell migration and infiltration.

This may or may not be true as ECM composition can vary between and within tumors and Matrigel is often considered to best represent the basal membrane composition rather than other types of intercellular stroma. Matrigel clearly has advantages for the growth of organoids but it may for example hinder T cell migration or activation. This sentence hence needs to be toned down quite a bit.

112: The model revealed targeting of organoids by all molecules, as evidenced by robust time- and concentration-dependent induction of apoptosis (Fig. 1e,f).

This sounds as if NT TCB had similar efficacy to the CEA and EPCAM targeted ones. I would recommend to revise.

Line 164: "TCBs targeting EpCAM, which is more accessible than CEA, triggered the strongest pattern of TCR downregulation, cytokine/GzmB expression, and cell cycling"

You describe before that you use CD3 downregulation (line 152) to assess T cell activation but now mention TCR downregulation. This is confusing.

Line 166: The sentence seems to be incomplete: "While CEA(hi) also induced a robust response, CD3 expression after CEA(lo) treatment remained constant, suggesting only modest TCR engagement triggered by healthy 23e, purple triangles.)"

Line 234: EpCAM TCB triggers proliferation of distant T cells in a contact-independent manner, which is possibly mediated by soluble cytokines diffusing away from the site of inflammation.

It may be due to cytokines but what has not been controlled for is the possibility that EPCAM TCB triggers proliferation even in the absence of target in this model. A no-organoid control is necessary to distinguish these two possibilities.

Fig 3: What are the dynamics of CD14 and CD20 cells? These should be analysed/quantified and briefly be discussed as well.

Line 399: In addition to identifying features that are predictive of patient response, we view this platform as a potential companion diagnostic in the clinic. Organoids and tumoroids can be rapidly generated from biopsies prior to treatment, and used for predicting anti-tumor efficacy and toxicity potential in the patient.

I am quite sceptical about this statement as organoids were generated from resection specimens, required several weeks of culture and the failure rate to establish organoids is not mentioned. Organoid culture requires quite demanding logistics, invasive procedures and time and I don't think the rapid generation from biopsies is a realistic description of this much more complex reality. In my view, the utility as a companion diagnostics is therefore rather limited.

Reviewer #3 (Report for the authors (Required)):

This study uses patient-derived (tumour) organoids to study both the efficacy and epithelial toxicity of bispecific T cell-engaging (Bite) antibodies. The authors adopt 3D organoid-PBMC co-cultures to perform fluorescent probe-enriched live cell imaging and quantitative multiplexed immunohistochemistry. These techniques allow for the (indirect) correlation of spatially annotated marker expression with functionally-defined outcomes, and are accompanied by thorough flow cytometry-assisted immunophenotyping data and cytokine assays. Altogether, this offers a solid and timely methodological advance. The report is also of interest as it addresses a clinical problem with Bite and CAR immunotherapy: the lack of predictive models for efficacy and/or toxicity, relevant for translational and clinical purposes. The platform can have a fairly strong impact not only for its intended purpose, but also for the field of tissue engineering and translational model-generation.

I have only moderate/minor remarks:

1. For the word 'mediated' in the subheaders on line 133 and 190: data only offer (strong) indirect support for claims of causal roles. I suggest softened the language.
2. Lines 146–149 and the figure S2a reference; it is confusing what data can be found where: the initial analysis of intracellular TNF α is not found in Fig S2a. Please clarify description data/figures. Also, the percentage notation in the y-axes in Fig S2a should be simplified.
3. Something went wrong in lines 167–168.
4. Fig 3c and Fig S3c are (nearly) identical, this is unnecessary: authors should remove one of them. Furthermore, in the interpretation of the data of CD8 vs CD4 T cells (lines 231–234), I find the logic not very strong: several other possibilities can account for the observation. Furthermore, the evidence for proliferation is far from clear in Fig S4b. Please provide close-ups and arrows to better appreciate proliferating T cells in their spatial context. Relatedly, this difference in penetration/accumulation is not seen in figure 4d
5. Lines 355–359: It is not very clear how this model or the data presented will guide optimization of CIT towards more favorable safety profiles. The CEA hi/lo example merely characterizes two forms with higher or lower activity in killing of both normal mucosa and tumoroids. The challenge is in teasing out larger therapeutic windows; it should be highlighted (discussed) how the platform will facilitate this.
6. On representation of image quantification data, I find the bars in Fig 3d not ideal: there is no way of telling the number of observations nor the data spread. Please consider box/violin plots and/or plotting individual data points. I understand it may not be possible to combine the caspase probe together with mIF analysis and, therefore, to correlate T cell infiltration to fate for an individual organoid/tumoroid; thus the premise that intra-glandular T cell infiltration 'appears to be necessary' for cytotoxicity sounds more appropriate than 'mediating'.
7. About lines 400–403. I am interested how the analysis of figure 5, but then extended to the matched tumoroids, would determine therapeutic windows, or rank the patients for predicted benefit. This would be a good proof-of-concept for the suggestion of this platform as a companion diagnostic. As it is, it seems like a missed opportunity.
8. Figure legend of Fig 2f: this would be intracellular expression, not 'secretion', right?
9. Fig 5f: x-axis label is missing.

Mon 14 Aug 2023

Decision on Article NBME-22-2306B

Dear Dr Gjorevski,

Thank you for your revised manuscript, "Organoid model to predict and study immunotherapy-induced damage of healthy and tumor tissue". Having consulted with the original reviewers (whose comments you will find at the end of this message), I am pleased to write that we shall be happy to publish the manuscript in *Nature Biomedical Engineering*, provided that the points of Reviewer #1 are addressed. In particular, your arguments on the difficulties in the use of allogeneic PBMCs should be clear from the manuscript or as supplementary discussion.

In the meantime, we will assess the main text in more detail, in particular the title and abstract, for clarity, accessibility and readability. Please expect an update to this message with additional points to address. However, you don't need to wait for the update to submit the final files.

For primary research originally submitted after December 1, 2019, we encourage authors to take up transparent peer review. If you are eligible and opt in to transparent peer review, we will publish, as a single supplementary file, all the reviewer comments for all the versions of the manuscript, your rebuttal letters, and the editorial decision letters. **If you opt in to transparent peer review, in the attached file please tick the box 'I wish to participate in transparent peer review'; if you prefer not to, please tick 'I do NOT wish to participate in transparent peer review'**. In the interest of confidentiality, we allow redactions to the rebuttal letters and to the reviewer comments. If you are concerned about the release of confidential data, please indicate what specific information you would like to have removed; we cannot incorporate redactions for any other reasons. More information on transparent peer review is available.

Best wishes,

Valeria

Dr Valeria Caprettini
Associate Editor, Nature Biomedical Engineering

P.S. Nature Portfolio journals encourage authors to share their step-by-step experimental protocols on a protocol-sharing platform of their choice. Nature Portfolio's Protocol Exchange is a free-to-use and open resource for protocols; protocols deposited in Protocol Exchange are citable and can be linked from the published article. More details can be found at www.nature.com/protocolexchange/about.

Reviewer #1 (Report for the authors (Required)):

The value of this co-culture system is that it can evaluate both efficacy of TBC against cancer cells and on-target off-tumor toxicity to normal cells. Otherwise, it is less original and indistinguishable from other existing co-culture systems except multiplexed immunofluorescence imaging. However, the imaging is not straightforward and complicate to interpret.

Even without TBC, allogenic PBMC attack to organoid and tumoroid as shown in fig. 1e. Therefore, prediction of on-target off-tumor toxicity of TBC is not reasonable using allogenic PBMC. Therefore, autologous PBMC co-culture experiment is crucial. In this revised manuscript, the authors showed data of two additional autologous PBMC experiments in supplemental figure 5. The efficacy to tumoroids is minimal comparing to allogenic PBMC and on-target off-tumor toxicity is similar.

Using autologous PBMC, comparison between no treat group and TBC treat group to organoids and tumoroids will potentiate this co-culture system. If there are technical limitation, the data of two additional autologous PBMC experiments should be presented as main data in the text

minor comment

Figure 1 legend"on target off tumor" In the last part of the sentence "toxicity" may be missed.

Reviewer #2 (Report for the authors (Required)):

The authors have made updates to the manuscript and performed additional experiments which have answered all my questions.

Reviewer #3 (Report for the authors (Required)):

My concerns have been sufficiently addressed.

Nature Biomedical Engineering is a Transformative Journal. Authors may publish their research with us through the traditional subscription access route, or make their paper immediately open access through payment of an article-processing charge. More information about publication options is available.

You may need to take specific actions to comply with funder and institutional open-access mandates. If the work described in the accepted manuscript is supported by a funder that requires immediate open access (as outlined, for example, by Plan S) and your manuscript was originally submitted on or after January 1st 2021, then you will need to select the gold OA route. Authors selecting subscription publication will need to accept our standard licensing terms (including our self-archiving policies), and these will supersede any other terms that the author or any third party may assert apply to any version of the manuscript.

Rebuttal 1

From: Gjorevski, Nikolche <nikolche.gjorevski@roche.com>

Sent: 08 October 2022 18:11

To: Valeria Caprettini <valeria.caprettini@nature.com>

Subject: Re: Decision on Article nBME-22-2306

[External - Use Caution]

Dear Dr. Caprettini,

I would like to thank you for taking the time to review our manuscript. Your feedback is much appreciated, especially bearing in mind the high number of submissions you must be receiving continuously. Although the rejection is disappointing, I fully respect your decision. Nonetheless, reading the specific reasons you cited for the rejection, I was hoping that I might kindly ask you to reconsider, based on the fact that these shortcomings can be readily addressed.

As main reasons for the outcome, you listed 1) previously published cancer mimicry/immunocompetent cancer organoids, 2) absence of evidence that the toxicity mechanisms in patients matched those observed within our models and 3) absence of molecular and functional characterization of the organoids. I would like to address each of these points individually:

1) Indeed, studies employing co-cultures of immune cells and tumor organoids have been widely published and we cite many of them in our manuscript. Our study, however, focuses on immunocompetent *healthy* organoids, in line with our aim to dissect toxicities of bispecific antibodies to healthy tissues, and specifically the intestine. We do utilize PBMC co-cultures with both tumor and healthy organoids as a means to evaluate the therapeutic index of the molecules, i.e. compare efficacy and safety effects. Our model correctly recapitulated the absence of a clinical therapeutic index of EpCAM TCB (Kebenko et al, *Oncoimmunol* 2018), unlike preclinical mouse models, which predicted a favorable therapeutic window (Amann et al, *Cancer Immunol Immunother* 2009).

Furthermore, whereas all previous tumor organoid studies evaluate the functional outcomes of the treatments using simplistic readouts (mainly apoptosis), we explore the underlying cellular mechanisms. Using multiplexed immunofluorescence (mIF) analysis, we describe the spatial and temporal changes within the immune cell compartment and its interactions with organoids that ultimately lead to cytotoxicity. The imaging resolution & quality required to generate these insights was directly enabled by performing mIF on thin histological sections of the co-cultures for the first time. This advance was, in turn, made possible by housing the co-cultures within solid ECM-like gels, unlike previous studies involving liquid-phase co-culture, which forces interactions between organoids and immune cells.

2) Insights into the cellular mechanisms of tissue toxicity associated with bispecifics are difficult to glean, because healthy tissues are simply not biopsied during the course of the treatment, which is when the adverse effects are at their most acute. In fact, our model can help understand -- and identify ways to treat -- this critical stage, which is currently a black box.

However, one patient treated with EpCAM TCB underwent endoscopic evaluation 7 days after treatment termination, owing to persistent abdominal pain (Kebenko et al, 2018). The histological findings are strikingly similar to the observations we derived from the organoid model:

- Damage to the crypt and villus structure of the patient duodenum was observed, which is consistent with epithelial apoptosis. Mucosal ulceration (complete erosion of the epithelium) was also noted, which is exactly what we find upon prolonged observation (>96 h after the start of treatment). These data are available.
- Kebenko et al report extensive lymphocyte infiltration into the tissue, including between epithelial cells. Our model recapitulates this effect (Fig. 3 and 4)
- Measurement of serum cytokines during infusion revealed increase in IFN γ , IL-6 and IL-8 levels. Our models replicates this finding also - these three cytokines are significantly induced in organoid-PBMC co-cultures in response to EpCAM TCB treatment (Fig. 2G).

3) We did not include extensive organoid characterization data (aside from expression of TCB targets EpCAM and CEA), because we considered intestinal organoids to be a rather standard and well-characterized system by now (Sato et al, *Gastroenterol* 2011; Fujii et al, *Cell Stem Cell* 2018; Lukonin et al, *Nature* 2020). However, we performed extensive characterization of our organoids using immunohistochemistry, molecular phenotyping and RNA sequencing. We would be happy to include these data into the manuscript.

In light of these points, would you be willing to reconsider your decision and allow the manuscript to be reviewed by external referees? Of course, we would ensure that a revised version of the manuscript addresses these points by additional discussion and added data.

Thank you very much for the further consideration.

Best wishes,
Nikolche

Rebuttal 2

We would like to thank all three reviewers for their thorough review and constructive comments. We believe that we have addressed their suggestions, thus increasing the clarity and potential for impact of our manuscript in the field.

Reviewer 1:

1. *“The authors described inter-donor variability of TCB toxicities, however, they did not describe inter-PBMC variability. To recapitulate clinical responses and toxicities, autologous PBMC and healthy / tumor organoids co-culture models from cancer patients seems more appropriate.”*

Upon the reviewer’s question and suggestion, we repeated the experiment (with a subset of 6 lines from the original 14 healthy organoid lines; with low- and high-CEA expressing donors being represented). Each organoid donor was co-cultured with three separate PBMC donors - all different from the PBMC donor included in the original manuscript.

Overall, we found that the outcomes were consistent with those reported in the original submission. In the repeat experiments, donors expressing higher CEA levels were targeted more extensively, and we derived a positive correlation between organoid apoptosis and target expression (Supp. Fig. 9a-e). The results were consistent across the three PBMC donors. The third PBMC donor was somewhat of an exception: whereas the overall inter-donor pattern of damage was preserved, the amount of damage seemed to be attenuated, especially in the case of the organoid donors with low CEA levels, which suggests that this PBMC donor might have a higher threshold of activation or is less reactive. Nonetheless, it is important to highlight that the same conclusions would have been drawn from all experiments, using all PBMC donors.

We also followed the reviewer’s suggestion and repeated the experiments using autologous organoid-tumoroid-PBMC combinations (Supp. Fig. 5). We would like to underscore that we experienced some technical challenges in addressing this point, mainly related to establishing additional lines in sufficient quantities and matched with enough autologous PBMCs. While our success rate for generating organoids from small specimens was nearly 100%, in many cases organoid and, especially, tumoroid growth was extremely slow. As a result, we only managed to expand 2 additional organoid-tumoroid pairs to the numbers required to perform revision experiments. The quantities of organoids/tumoroids and, particularly PBMCs, then dictated the experimental design and the number of conditions we could include. Although we were able to isolate a sufficient quantity of PBMCs from both donors, we observed low viability and, hence, low reactivity of those originating from donor p634. Nonetheless, we were able to investigate EpCAM TCB-triggered apoptosis and T-cell infiltration in both organoid and tumoroid culture. Furthermore, we compared the outcomes between co-cultures utilizing allogeneic and autologous PBMCs. Below, we outline the main findings:

- In both additional organoid-tumoroid pairs, EpCAM TCB induced significantly more apoptosis and T-cell infiltration in organoids, compared with tumoroids (Supp. Fig. 5a-f), which is well in line with our original findings
- Comparing outcomes of experiments performed with allogeneic vs autologous PBMCs, we obtained similar results in the case of healthy organoids, i.e. allogeneic and autologous PBMCs induced comparable levels of apoptosis upon TCB treatment.
- In contrast, we observed that allogeneic PBMCs targeted tumoroids more extensively than autologous ones. It is possible that any neoantigen-driven (bystander) T-cell activation is further amplified by an alloreactive response.

Although it would be tempting to recommend that it is permissible to use allogeneic PBMCs in organoid co-cultures, whereas they may overestimate targeting efficacy of tumoroids, we should also bear in mind that we compared the outcome to a single allogeneic PBMC donor. As discussed above, PBMC donors do have slight differences in reactivity, and we believe that more work is needed to be able to make definitive claims. It is also crucial to highlight again that the overall conclusions of the experiments are consistent, regardless of the PBMC donors or whether they are matched or non-matched to the organoids and tumoroids: our model in all cases captures TCB-induced toxicities in the form of organoid apoptosis, which is associated with T-cell infiltration and the extent of which is mainly dictated by target expression. Finally,

all experiments we performed confirm that tumoroids are more resistant to damage, in line with failures to reach therapeutic doses of these molecules in the clinic.

2. *“A limitation of the co-culture model is not applicable for predicting off-tumor toxicities of immune checkpoint inhibitors. Limitations of this models for predicting off-tumor toxicities of T-cell targeted approach should be mentioned”*

We agree with the reviewer that our co-culture model is likely not appropriate for predicting intestinal toxicities associated with immune checkpoint inhibitors (ICI), owing to their complex and poorly understood mechanisms, which may involve not only T cells but also myeloid and stromal cells, within the tissue or within lymphoid organs (Sullivan et al, *Nat Rev Drug Dis* 2022). We have added text that clearly communicates this limitation. We also reference an elegant recent study by the Wucherpennig laboratory which we find pertinent to the topic, as it dissects the mechanisms of ICI-induced colitis (Luoma et al, *Cell* 2020). Using scRNA-seq analysis of primary patient samples, Luoma and co-authors find that ICI-triggered colitis is principally driven by antigen-experienced, tissue-resident populations. These results are encouraging, as they suggest that ICI toxicities may also be captured by rather simple organoid-immune cell co-cultures, without having to rebuild the entire antigen-presentation and recirculation machinery *in vitro*. In this case, however, organoids would have to be co-cultured not with PBMCs but rather with intestine-derived tissue-resident memory T cells, which should theoretically be retrievable from biopsies and resections.

3. *“Unlikely to healthy organoids, or tumoroids morphology is not uniform as described in other reports. In this manuscript, the authors did not mention the number of combined safety-efficacy experiments. Considering intertumoral heterogeneity, several additional experiments using tumoroids of other morphology potentiate the claim of combined safety-efficacy profiling platform.”*

We thank the reviewer for the suggestion. We also were curious about whether our findings could be extended to additional organoid-tumor pairs, as the combined safety-efficacy experiments were indeed performed using a single donor in the original manuscript. To address the question, we established two additional organoid-tumoroid pairs, as already discussed above (Supp. Fig. 5) and obtained similar results as before.

We wished to push further and also address the reviewer’s question regarding the influence of tumoroid morphology on the TCB-induced toxicities analyzed in this study. We established additional tumoroid lines until we had a set of three with variable morphologies (Supp. Fig. 7a). In particular, one tumoroid line featured a simple columnar epithelium and a large lumen, closely resembling healthy organoids. The second line featured a packed morphology and nearly lacked a lumen, whereas the third line’s morphology fell between that of the other two: it combined packed, multilayered regions with simple, lumenized ones. We performed the co-culture experiment, choosing to treat with EpCAM TCB, being mindful that target expression may confound the effect of morphology, and EpCAM expression is less variable than that of CEA. Indeed, we confirmed that all lines expressed EpCAM robustly and at similar levels, although subtle differences were indeed observed. Despite the striking differences in morphology, we did not observe major differences in outcome: all three lines were targeted by the TCB, and underwent T-cell infiltration and ultimately apoptosis (Supp. Fig. 7b-d). Interestingly, the subtle differences in both organoid killing and T-cell entry were explained by differences in EpCAM expression: despite having an intermediate morphology, line p471 was most extensively infiltrated and killed, likely owing to the highest amount of EpCAM present (Supp. Fig. 7e-g). Therefore, these data confirm our findings that target expression is the most robust determinant of tumoroid susceptibility to TCB-mediated damage.

4. *“Inter-personal variability may influence growth of organoids and eventually affect to organoid differentiation, morphology and target expression. As shown in figure 1b and figure 4a, individual organoids in a single culture also differ in target expression. If TCB toxicities and efficacy are related to amount of target expression, mIF can visualize more T-cell infiltration in organoids with higher target expression than low target expression in a single experiment”*

The review is correct in her/his statement. We repeated the multi-donor experiment described in Fig. 5 and, in addition to live imaging of caspase induction, we fixed & sectioned the co-cultures and performed mIF to analyze T-cell infiltration. Whereas powerful in providing deep mechanistic insights, the mIF approach is not high-throughput, so we focused on a subset of 6 donors to maintain the experiment practically manageable. As the reviewer anticipated, we indeed observe that organoid lines which express high levels of CEA (Supp. Fig. 9c) and undergo most apoptosis (Supp. Fig. 9a, b, d) (i.e. donors 8 and 13), also experience the greatest extent of both CD4 and CD8 T-cell infiltration (Supp. Fig. 9e). Overall, we find strong positive correlation between target expression and T-cell integration (Supp. Fig. 9f).

5. *"In 96-well plate seeding, 15,000 PBMCs and an unknown number of organoids are co-cultured in a 5ul Matrigel dome/well. If the number of cells is greater than a proper amount of Matrigel, there may be difficulty in Matrigel hardening and a risk of dome detachment from culture plate during media change and culturing. Precise description regarding the ratio of organoids/PBMC cell number and total cell number are required"*

We apologize for the lack of details on the experimental protocol. We have now added this information in the Methods section. Number of organoids per dome varies with amount of matrix used. ~40 organoids per well for the killing assay (10 μ L domes) determined to yield the best results, both in terms of organoid segmentation during image analysis as well as statistical power, whereas ~100 organoids were seeded when performing the FFPE-based imaging approach in a 25 μ L domes. Exact organoid to PBMC cell number was confirmed by mIF quantification across dozens of samples at baseline (0 h condition). Analysis of the images yielded a 1.1:1 of E:T ratio (considering EpCAM+ cells as the entire PBMC population). Only considering the main effectors (T cells), the ratio results in a 1:2.5 E:T ratio. Given the 10 μ L dome for the killing assay, this results in 150000 PBMCs to roughly 136000 epithelial cells. Until now, difficulty in Matrigel hardening has never been observed.

6. *"Due to the sticky nature of Matrigel, technical problem such as air bubble formation and phase separation easily occur during resuspension process. For this reason, mentioning of success rate and total number of co-culture and TCB treatment experiments are required."*

As we had alluded to above, the approach we describe is primarily limited by the availability (quantity and quality) of the starting material – we must have viable organoids and PBMCs in sufficient numbers. The technical aspects of the protocol itself (including the resuspension) are straightforward to an experienced experimenter. We have performed more than a 100 of these experiments, comprising more than 1000 co-cultures. We have "lost" only a few conditions due to technical issues like bubbles or gel detachment, accounting for a success rate of more than 95% in our estimation.

7. *"Display of data in Fig 5 is complicating. Sequences of donor organoids number in Fig 5a, b, c, d is inconsistent"*

We thank the reviewer for pointing this out. Our reasoning behind the inconsistent sequences is the following: the number order in Fig. 5a is random, because at the start of the study, we had no parameter to guide the ordering. In Fig. 5b, the organoid lines are ordered by extent of apoptosis experienced, whereas in Fig. 5c,d they are ordered by target expression. We now state the ordering rationale in the figure legend to help guide the reader.

8. *"Lane #246 Fig. 3a is not matched with the text"*

We thank the reviewer for pointing out this mistake. We have now corrected it.

Reviewer 2:

1. *"Harter et al describe the use of human intestinal epithelial organoids from normal small and large bowel mucosa as in vitro models to assess the toxicity of immunotherapies that target cell surface"*

antigens. This model is novel and highly important as animal models that recapitulate the expression of relevant human antigens are often lacking. This is hence a major new addition to the repertoire of tests that can be used to define potential safety issues with new immunotherapeutics before testing these in patients”

We thank the reviewer for recognizing the novelty and potential impact of our work.

2. “Line 286: “high ZO-1 expression might be considered an immune-protective, whereas E287 cadherin expression an immune-inclusive feature”

I think this is too simplistic. ZO-1 expression needs to be in the right place to hinder infiltration and E-cadherin is not predominantly an immune regulatory molecule but has a broad range of functions. Rephrasing would be useful.

Figure 4f, g: the staining pattern further confirms my concern above. ZO-1 is for example mainly expressed on the luminal side of the organoids where it is unlikely to impair T cell infiltration. And although the optical density of E cadherin is numerically lower in cancer cells, it still stains strongly in these, questioning whether this difference really has a relevant impact. I think you would need blocking experiments to assess whether E cadherin expression is important for infiltration. It is also not clear how many different organoid-tumoroid pairs were used in these experiments. In order to make general statements about differences in organoid and tumoroid, this comparison needs to be done in a good number of these. The same applies for the statement “Given the striking differences in epithelial architecture and morphology between the two tissues types (Fig. 4a),”. You need to show this across a reasonable number of organoid-tumoroid pairs”

We agree with the reviewer that the data presented in the original submission did not fully substantiate our claims regarding the differential expression of junction molecules driving the differential immune cell infiltration in healthy and tumor epithelium. We also agree that the model is too simplistic and have therefore decided to fully remove these claims, especially in light of the additional data we generated over the past several months.

As the reviewer suggested, we used antibodies to block the interaction between lymphocytes and E-cadherin, which has been shown to proceed via the CD103 receptor on the former, upon their activation (Rogoz et al, *J Immunol Meth* 2015; Morikawa et al, *CMGH* 2021). To our surprise, the blocking experiments resulted in neither decreased organoid targeting nor attenuated T-cell infiltration within the organoids (Supp. Fig. 6c-e). Puzzled by this result, we checked CD103 expression within T-cells, which we presumed to be induced upon activation, as shown before (Supp. Fig. 6f, g). We found that TCB-mediated T-cell activation and organoid targeting is not associated with CD103 upregulation, suggesting that lymphocytes infiltrate the organoid epithelium via a mechanism different from the classical CD103-E-cad interaction. Bearing in mind that TCB targets are surface molecules, we believe that, in the context of the toxicities described here, lymphocyte epithelial infiltration is driven by CD3-target (CEA/EpCAM) interaction. To address this hypothesis, we repeated the multi-donor experiment described in Fig. 5 and, in addition to live imaging of caspase induction, we fixed & sectioned the co-cultures and performed mIF to analyze T-cell infiltration. Whereas powerful in providing deep mechanistic insights, the mIF approach is not high-throughput, so we focused on a subset of 6 donors to maintain the experiment practically manageable. We indeed observe that organoid lines that express high levels of CEA (Supp. Fig. 3g, h) (i.e. donors 8 and 13), also experience the greatest extent of both CD4 and CD8 T-cell infiltration (Supp. Fig. 9e). Overall, we find positive correlation between target expression and T-cell integration (Supp. Fig. 9f).

Next, we addressed the reviewer’s further comment on extending the comparisons of organoid vs. tumoroid susceptibility to damage and infiltration, by generating two additional organoid-tumoroid pairs

(Supp. Fig. 5a). Consistently with the findings in the original submission, EpCAM TCB triggered higher apoptosis and intraepithelial integration of T cells within tumoroids, compared with organoids (Supp. Fig. 5b-f). Of note, we performed co-culture experiments with both allogeneic and autologous PBMCs. One of the autologous PBMC donors (p634) appeared to be poorly reactive, which was consistent with the low cell viability we recorded prior to the experiment. The other autologous PBMC donor (p645) performed as expected, damaging and infiltrating tumoroids more than organoids (Supp. Fig. 5c-f).

We also wished to further test our claim that differences in morphology *per se* may drive the differences in T-cell infiltration and organoid killing. We established additional tumoroid lines until we had a set of three with variable morphologies (Supp. Fig. 7a). In particular, one tumoroid line featured a simple columnar epithelium and a large lumen, closely resembling healthy organoids. The second line featured a packed morphology and nearly lacked a lumen, whereas the third line's morphology fell between that of the other two: it combined packed, multilayered regions with simple, lumenized ones. We performed the co-culture experiment, choosing to treat with EpCAM TCB (Supp. Fig. 7a-c), being mindful that target expression may confound the effect of morphology, and EpCAM expression is less variable than that of CEA. Indeed, we confirmed that all lines expressed EpCAM robustly and at similar levels, although subtle differences were indeed observed. Despite the striking differences in morphology, we did not observe major differences in outcome: all three lines were targeted by the TCB, and underwent T-cell infiltration and ultimately apoptosis (Supp. Fig. 7d-e). Interestingly, the subtle differences in both organoid killing and T-cell entry were explained by differences in EpCAM expression: despite having an intermediate morphology, line p471 was most extensively infiltrated and killed, likely owing to the highest amount of EpCAM present (Supp. Fig. 7c). Therefore, these data confirm our findings that target expression is the most robust determinant of organoid susceptibility to TCB-mediated damage, rather than morphology.

These new data explain that differences in lymphocyte infiltration between different tumor- or healthy organoid donor lines are largely driven by differences in target expression. We recognize, however, that we still do not understand what drives the differences in infiltration between tumoroids and organoids of the same donor. We find this question to be fascinating, but also recognize that it warrants a separate in-depth study that may reveal interesting fundamental differences in tumor vs healthy epithelium that may be leveraged therapeutically.

3. *"Line 299: Quantification of TCB-mediated organoid killing revealed striking differences in response within the organoid cohort (Fig. 5a,b), wherein certain organoid lines were targeted readily and others seemed refractory to damage.*

A key question here is how many different donors were used for the PBMC generation and whether the striking differences you mention were observed in experiments done at the same time and with the same donor. How many repeats were performed? The lack of this kind of detailed information makes the rigidity of the experimental approach difficult to judge".

We thank the reviewer for raising this important point and we apologize for the lack of clarity in the original submission. Given the scale, the original experiment involving multiple organoid donors was performed once. All organoid donors were included in the same experiment and tested simultaneously, to minimize differences owing to technical/experimental variation. A single PBMC donor was used. The 14-organoid line/1-PBMC donor experiment was repeated twice.

Upon the reviewer's question and suggestion, we repeated the experiment with a subset of 6 lines from the original 14, with low- and high-CEA expressing donors being represented. Each organoid donor was co-cultured with three separate PBMC donors (Supp. Fig. 9). The 6-organoid line/3-PBMC line experiment was then repeated twice.

Overall, we found that the outcomes were consistent with those reported in the original submission. In the repeat experiments, donors expressing higher CEA levels were targeted more extensively, and we derived a positive correlation between organoid apoptosis and target expression (Supp. Fig. 9f). The results were

consistent across the three PBMC donors. The third PBMC donor was somewhat of an exception: whereas the overall inter-donor pattern of damage was preserved, the amount of damage seemed to be attenuated, especially in the case of the organoid donors with low CEA levels, which suggests that this PBMC donor might have a higher threshold of activation or is less reactive. Nonetheless, it is important to highlight that the same conclusions would have been drawn from all experiments, using all PBMC donors.

4. *“When discussing results from Fig 5, a more quantitative approach would be useful as the description is rather vague as it stands. What proportion of organoids showed what degree of damage?”*

We apologize for the lack of clarity with the killing readouts, as reported in the original submission. To derive a more tangible readout of the physical outcomes associated with TCB treatment, we used the mIF approach. Specifically, we quantified the areas of organoids expressing caspase (dead) vs expressing E-cadherin (live) and reported the ratio across the different donors treated with TCB for 48 h (Supp. Fig. 3h). The plot shows the degree of damage incurred by individual organoids. The outcomes were consistent across different organoids, all organoids showed similar amounts (or lack) of damage.

5. *“In the final paragraph of the results section, I would have thought that it makes sense to assess reasons for different sensitivities in more detail, for example with by performing the same FACS analyses (as shown in Fig 2) to assess T cell activation status and effector functions and by 3D multiplex immunostaining (as shown in Fig 3) to investigate whether this is due to a lack of T cell infiltration.”*

We thank the reviewer for this suggestion. As we mentioned earlier, the mIF approach is not a high-throughput readout, so we hoped that repeating the experiment with a subset of 6 organoid lines would nonetheless answer the question. We indeed found that organoid lines that were heavily damaged were also most extensively infiltrated by both CD4⁺ and CD8⁺ T cells (Supp. Fig. 9e, f). To assess the T-cell activation status, we used multiplex ELISA for proinflammatory cytokines released in the supernatant. Consistently, we found highest induction of cytokines in lines 8, 9 and 13, which 1) express high levels of target, 2) undergo most extensive damage and 3) are most infiltrated by immune cells (Supp. Fig. 9g).

6. *“Discussion: You show that small intestine and colon organoids differ in EPCAM and CEA expression but it is not mentioned whether this now correlates with small or large bowel toxicities that were observed for such therapies in the clinic. I did CEA targeting drugs mainly trigger large bowel toxicities and does this match observations in your models? I am not sure how much of this clinical trial data has been published but the issue deserved at least some discussion.”*

Having this information would be a very relevant way to validate the findings from our model. However, patients who were treated with EpCAM- or CEA-targeted therapies and experienced gastrointestinal adverse events unfortunately did not undergo closer endoscopic examination (Pishvaian, *Clin Colorectal Cancer* 2016; Kebenko et al, *Oncoimmunology* 2018) – treatment was simply terminated. Broad intestinal toxicity was deduced from symptoms of abdominal pain and diarrhea, making it challenging for the investigators to localize the effects to a specific segment of the intestine. We actually believe that our model could complement clinical studies in providing additional mechanistic insights that would be difficult to obtain in the clinical setting, because of practical or medical reasons. We have now added a sentence in the discussion to clarify that current clinical studies do not provide information on TCB effects on different intestinal regions.

7. *“I understand that the surgical specimens were obtained from a commercial provider but the methods section should nevertheless provide information on how ethics approval for their use was obtained and whether patients provided written informed consent.”*

We have now added the missing information and thank the reviewer for pointing out the omission.

8. *“Line 52: The authors should also cite Gonzalez-Exposito JITC 2019 and Schnalzger Embo J 2019 as these were the first manuscripts showing the feasibility of 3D organoid + T cell co-culture to assess novel antigen specific therapeutics in bowel cancers.”*

We have now cited these important studies.

9. "A table showing the number of organoids, the anatomical origin, and basic donor characteristics (age, sex) should be included in the supplements."

We thank the reviewer for the suggestion. We have now included such a table in the Methods section.

10. "111: The solid extracellular matrix (ECM) surrounding the organoids and PBMCs is a more faithful representation of the native tissue context and allows simulating crucial immunological processes beyond contact-dependent targeting, including bystander signaling, immune cell migration and infiltration.

This may or may not be true as ECM composition can vary between and within tumors and Matrigel is often considered to best represent the basal membrane composition rather than other types of intercellular stroma. Matrigel clearly has advantages for the growth of organoids but it may for example hinder T cell migration or activation. This sentence hence needs to be toned down quite a bit."

We apologize for the lack of clarity. With this sentence, we meant to communicate that a solid 3D matrix (rather than the suspension-based or 2D approaches used in previous studies) is a more faithful representation of the tissue environment within solid tissue (both healthy and tumor). We referred to the mechanical and 3D format of the matrix, rather than the specific protein composition. We have now modified this part of the manuscript to more effectively communicate this point.

11. "112: The model revealed targeting of organoids by all molecules, as evidenced by robust time- and concentration-dependent induction of apoptosis (Fig. 1e,f).

This sounds as if NT TCB had similar efficacy to the CEA and EPCAM targeted ones. I would recommend to revise."

We thank the reviewer for pointing this out – it is indeed a confusing statement and we have now modified it to communicate that only the targeted TCBs had effects on the organoids and tumoroids.

12. "Line 164: "TCBs targeting EpCAM, which is more accessible than CEA, triggered the strongest pattern of TCR downregulation, cytokine/GzmB expression, and cell cycling"

You describe before that you use CD3 downregulation (line 152) to assess T cell activation but now mention TCR downregulation. This is confusing."

We apologize for the confusion. We used TCR and CD3 interchangeably because the TCR:CD3 complex is down-modulated during T-cell activation (through intracellular retention). In the study, we measure cell-surface expression of CD3 as a means to assess the levels of the CD3-TCR complex, and, hence, activation. We have made this concept clearer in the revised manuscript.

13. "Line 166: The sentence seems to be incomplete: "While CEA(hi) also induced a robust response, CD3 expression after CEA(lo) treatment remained constant, suggesting only modest TCR engagement triggered by healthy 23e, purple triangles)."

We thank the reviewer for pointing out this typo. We have now corrected it.

14. "Line 234: EpCAM TCB triggers proliferation of distant T cells in a contact-independent manner, which is possibly mediated by soluble cytokines diffusing away from the site of inflammation.

It may be due to cytokines but what has not been controlled for is the possibility that EPCAM TCB triggers proliferation even in the absence of target in this model. A no-organoid control is necessary to distinguish these two possibilities"

In all experiments reported in the manuscript, we include a non-targeting TCB (NT TCB), which contains a CD3-binding arm, but lacks an epithelial target-binding arm. Therefore, the antibody can bind T cells, but

not organoids and effectively represents a no-organoid control. In the experiment referenced by the reviewer, we see no effects in the NT TCB condition (despite engagement of the CD3 arm of T cells), which led us to the conclusion that the effects in the EpCAM TCB-treated co-culture are mediated by the combined organoid-T-cell engagement, rather than activation of T cells alone.

15. *“Fig 3: What are the dynamics of CD14 and CD20 cells? These should be analysed/quantified and briefly be discussed as well.”*

We indeed analyzed and quantified the distributions of CD14 and CD20 cells over time, in response to TCB treatment. Although these cells are not directly targeted by the TCBs, we considered that they may be affected via the inflammatory cytokines and chemokines produced in the process. However, the mIF analysis revealed no substantial changes in either their numbers or spatial localization. We have now communicated this in the revised version of the manuscript.

16. *“Line 399: In addition to identifying features that are predictive of patient response, we view this platform as a potential companion diagnostic in the clinic. Organoids and tumoroids can be rapidly generated from biopsies prior to treatment, and used for predicting anti-tumor efficacy and toxicity potential in the patient.*

I am quite sceptical about this statement as organoids were generated from resection specimens, required several weeks of culture and the failure rate to establish organoids is not mentioned. Organoid culture requires quite demanding logistics, invasive procedures and time and I don't think the rapid generation from biopsies is a realistic description of this much more complex reality. In my view, the utility as a companion diagnostics is therefore rather limited.”

We thank the reviewer for raising this important point. We had first-hand experiences of the limitations brought up by the reviewer, when trying to generate additional organoid and tumoroid lines for this revision under time pressure. While our success rate for generating organoids from small specimens was nearly 100%, in many cases (in fact, for 5 out of the 10 lines we attempted to generate), organoid and particularly tumoroid growth was extremely slow. As a result, we only managed to expand 2 additional organoid-tumoroid pairs to the numbers required to perform revision experiments. In light of this, we have toned down the statement on the utility of the assay as a companion diagnostic. We have also referenced interesting new approaches that are based on droplet microfluidics, which claim to expedite and increase the efficiency of organoid formation from biopsies (Ding et al, *Cell Stem Cell* 2022). These advances could help render the clinical application of organoid-based models feasible in the future.

Reviewer 3:

1. *“This study uses patient-derived (tumour) organoids to study both the efficacy and epithelial toxicity of bispecific T cell-engaging (Bite) antibodies. The authors adopt 3D organoid–PBMC co-cultures to perform fluorescent probe-enriched live cell imaging and quantitative multiplexed immunohistochemistry. These techniques allow for the (indirect) correlation of spatially annotated marker expression with functionally-defined outcomes, and are accompanied by thorough flow cytometry-assisted immunophenotyping data and cytokine assays. Altogether, this offers a solid and timely methodological advance. The report is also of interest as it addresses a clinical problem with Bite and CAR immunotherapy: the lack of predictive models for efficacy and/or toxicity, relevant for translational and clinical purposes. The platform can have a fairly strong impact not only for its intended purpose, but also for the field of tissue engineering and translational model-generation.”*

We thank the reviewer for the encouragement, and for recognizing the novelty and potential impact of our work

2. *“For the word 'mediated' in the subheaders on line 133 and 190: data only offer (strong) indirect support for claims of causal roles. I suggest softened the language.”*

We agree with the reviewer and have now changed the word 'mediated' to 'associated with'.

3. *"Something went wrong in lines 167–168."*

We thank the reviewer for pointing out the typographic issue. We have now corrected it.

4. *"Lines 146–149 and the figure S2a reference; it is confusing what data can be found where: the initial analysis of intracellular TNF α is not found in Fig S2a. Please clarify description data/figures. Also, the percentage notation in the y-axes in Fig S2a should be simplified."*

We thank the reviewer for pointing out this omission. We have now added a graph showing that modulation of TNF α is most pronounced in the GzmB+ CD8+ T cells and CD45RO+ memory CD4+ T cells.

The notation in the y-axes indicates the percentage occupied by the cell type in the plot from the collective CD45+ immune cell pool analyzed. We believe that the readers will find it clear, but are also open to suggestions on more effective labeling.

5. *"Fig 3c and Fig S3c are (nearly) identical, this is unnecessary: authors should remove one of them. Furthermore, in the interpretation of the data of CD8 vs CD4 T cells (lines 231–234), I find the logic not very strong: several other possibilities can account for the observation. Furthermore, the evidence for proliferation is far from clear in Fig S4b. Please provide close-ups and arrows to better appreciate proliferating T cells in their spatial context. Relatedly, this difference in penetration/accumulation is not seen in figure 4d"*

We apologize for the duplication and have changed the panel in FigS3c. We agree, the previous images in Fig S4B of the original submission were not really clear. We now enlarged the images, highlighting both GzmB and Ki67 in distinct panels together with the co-localizing CD4 and CD8 T cells. We have also added magnified inserts to highlight the proliferative state as well as GzmB degranulation of the T lymphocytes upon EpCAM TCB and compared it to NT TCB. Nonetheless, we also toned down our claim, bearing in mind that multiple mechanisms could account for the outcome.

The heatmap in Fig. 4 shows increased infiltration of CD8 T cells in organoids vs tumoroids upon both EpCAM TCB and CEA(Hi) TCB treatment and an increased CD4 T-cell infiltration in organoids upon CEA(Hi) TCB treatment (darker blue tones). We do agree with the reviewer, however, that the heat map representation may render these differences difficult to discern. We extended these experiments to two additional organoid-tumoroid pairs, confirming the higher T-cell infiltration in the latter. We have now shown these data as dot plots (Supp. Fig. 5e, f).

6. *"Lines 355–359: It is not very clear how this model or the data presented will guide optimization of CIT towards more favorable safety profiles. The CEA hi/lo example merely characterizes two forms with higher or lower activity in killing of both normal mucosa and tumoroids. The challenge is in teasing out larger therapeutic windows; it should be highlighted (discussed) how the platform will facilitate this."*

We agree with the reviewer that lowering the affinity of a typical antibody would shift the therapeutic window, rather than expanding it. However, recent TCB designs circumvent that problem. In these cases, lowering the antibody affinity has been shown to maintain anti-tumor activity in mouse models and patients (Bacac et al, *Clin Cancer Res* 2016; Tabernaro et al, *J Clin Onc* 2017), at doses that are well tolerated, owing to the avidity effect of these molecules. More specifically, the outcome is a result of the 2+1 format of this particular molecule (CEA TCB), whereby the antibody binds T cells monovalently, but binds the epithelial target bivalently, effectively better distinguishing between low- and high-target expressing cells and thus ensuring a non-linear gain in efficacy against tumor cells which express more CEA (Bacac et al, *Clin Cancer Res* 2016; Bacac et al, *Oncoimmunology* 2016).

7. *"On representation of image quantification data, I find the bars in Fig 3d not ideal: there is no way of telling the number of observations nor the data spread. Please consider box/violin plots and/or plotting individual data points. I understand it may not be possible to combine the caspase probe together with*

mIF analysis and, therefore, to correlate T cell infiltration to fate for an individual organoid/tumoroid; thus the premise that intra-glandular T cell infiltration ‘appears to be necessary’ for cytotoxicity sounds more appropriate than ‘mediating’.”

We thank the reviewer for pointing this out. We have now included a version of Fig. 3d that quantifies T-cell infiltration per organoid (combined Zone 1 and Zone 2 of the original plot) across the entire organoid population in the sample (Supp. Fig. 3i), thus providing a measure of the spread and variability of the readout.

To the reviewer’s second point. This is correct – we are currently unable to correlate the extent of damage of a particular organoid to the extent of infiltration at a previous time point. Therefore, we have followed the reviewer’s suggestion and changed the wording to ‘appears to be necessary’.

8. *“About lines 400–403. I am interested how the analysis of figure 5, but then extended to the matched tumoroids, would determine therapeutic windows, or rank the patients for predicted benefit. This would be a good proof-of-concept for the suggestion of this platform as a companion diagnostic. As it is, it seems like a missed opportunity.”*

We agree with the reviewer that exploring inter-donor variability of the therapeutic window would be fascinating. However, the 14-donor experiment was already a huge endeavor for us logistically, in terms of deriving the healthy organoid lines and performing the co-culture experiments. Doubling the experiment by introducing matched tumoroid lines would likely render the experiment unfeasible without the support from a larger team and/or investing in automation of the co-cultures and analysis.

We should point out that we have softened and qualified our claims of the potential of this platform as a companion diagnostic, based on the input from another reviewer. The current success rate and especially timescales required to establish organoids and tumoroids from small surgical samples likely render the application of organoid-based models as companion diagnostic impractical. We have, however, referenced interesting new approaches that are based on droplet microfluidics, which claim to expedite and increase the efficiency of organoid formation from biopsies (Ding et al, *Cell Stem Cell* 2022). These advances could help increase the feasibility of organoid-based companion assays in the clinic.

9. *“Figure legend of Fig 2f: this would be intracellular expression, not ‘secretion’, right?”*

The reviewer is correct and we thank her/him for pointing out this mistake. We have now corrected it

10. *“Fig 5f: x-axis label is missing.”*

We thank the reviewer for catching the omission. We have now added the x-axis label.

Rebuttal 3

Reviewer 1:

1. "The value of this co-culture system is that it can evaluate both efficacy of TBC against cancer cells and on-target off-tumor toxicity to normal cells. Otherwise, it is less original and indistinguishable from other existing co-culture systems except multiplexed immunofluorescence imaging. However, the imaging is not straightforward and complicate to interpret.

Even without TBC, allogenic PBMC attack to organoid and tumoroid as shown in fig. 1e. Therefore, prediction of on-target off-tumor toxicity of TBC is not reasonable using allogenic PBMC. Therefore, autologous PBMC co-culture experiment is crucial. In this revised manuscript, the authors showed data of two additional autologous PBMC experiments in supplemental figure 5. The efficacy to tumoroids is minimal comparing to allogenic PBMC and on-target off-tumor toxicity is similar.

Using autologous PBMC, comparison between no treat group and TBC treat group to organoids and tumoroids will potentiate this co-culture system. If there are technical limitation, the data of two additional autologous PBMC experiments should be presented as main data in the text"

We would like to thank the reviewer for carefully reading the revised manuscript and providing feedback.

To the reviewer's first point, we would like to respectfully point out that the green fluorescence signal in Fig. 1e is the result of homeostatic shedding of the organoids, which contain terminally differentiated cells that undergo turnover. The apoptosis signal associated with TCB-induced T-cell activation and killing is significantly higher, as shown in the quantification (Fig. 1f). This is further supported by high-resolution mIF images of control co-cultures (treated with non-targeting TCB for 72 h) (shown below and in Fig. 3b), indicating no damage to the epithelium due to allo-reactivity alone. Immuno-profiling by flow cytometry (Fig. 2) likewise shows no evidence of treatment-independent activation of PBMCs within the co-cultures, even after 72 h. We therefore conclude that no organoid targeting owing to allo-reaction occurs in our co-culture system within the experimental window.

With regards to the question of using autologous vs. allogenic PBMCs in the assay, we find that both options have advantages and disadvantages. As the reviewer pointed out (and as we mention ourselves) we observe enhanced killing of tumor organoids when allogenic PBMCs are used, which may be the consequence of additional immune cell activation owing to alloreactivity. However, it could also be the result of the typically lower viability and quality of autologous PBMCs collected from cancer patients compared with commercially sourced cells. Indeed, the window in TCB-mediated killing of healthy vs tumor organoids would differ depending on whether the co-culture was performed using autologous or allogenic PBMCs. We would be cautious to draw this conclusion based on one experiment, however, and the question requires further investigation. Nonetheless, all experiments yielded the same results regarding the main TCB effects: 1) whether healthy organoids are targeted and 2) whether tumor organoids are targeted more or less in comparison; that is, whether a therapeutic window exists. Therefore, our assay can be reliably used to answer these questions with allogenic PBMCs, even though

it might not be able to provide an exact therapeutic window and TCB concentrations. We have now clearly communicated these concepts and limitations as additional discussion.